# Brown adipose tissue monocytes support tissue expansion

Alexandre Gallerand[1,10], Marion I. Stunault[1,10], Johanna Merlin[1], Hannah P. Luehmann[2], Deborah H. Sultan[2], Maria M. Firulyova[3], Virginie Magnone[4], Narges Khedher[1], Antoine Jalil[5], Bastien Dolfi[1], Alexia Castiglione[1], Adelie Dumont[1], Marion Ayrault[1], Nathalie Vaillant[1], Jérôme Gilleron[1], Pascal Barbry[4], David Dombrowicz[6], Matthias Mack[7], David Masson[5], Thomas Bertero[4], Burkhard Becher[8], Jesse W. Williams[9], Konstantin Zaitsev[3], Yongjian Liu[2], Rodolphe R. Guinamard[1], Laurent Yvan-Charvet[1] & Stoyan Ivanov[1✉]

Monocytes are part of the mononuclear phagocytic system. Monocytes play a central role during inflammatory conditions and a better understanding of their dynamics might open therapeutic opportunities. In the present study, we focused on the characterization and impact of monocytes on brown adipose tissue (BAT) functions during tissue remodeling. Single-cell RNA sequencing analysis of BAT immune cells uncovered a large diversity in monocyte and macrophage populations. Fate-mapping experiments demonstrated that the BAT macrophage pool requires constant replenishment from monocytes. Using a genetic model of BAT expansion, we found that brown fat monocyte numbers were selectively increased in this scenario. This observation was confirmed using a CCR2-binding radiotracer and positron emission tomography. Importantly, in line with their tissue recruitment, blood monocyte counts were decreased while bone marrow hematopoiesis was not affected. Monocyte depletion prevented brown adipose tissue expansion and altered its architecture. Podoplanin engagement is strictly required for BAT expansion. Together, these data redefine the diversity of immune cells in the BAT and emphasize the role of monocyte recruitment for tissue remodeling.

[1] Université Côte d'Azur, INSERM, C3M, Nice, France. [2] Department of Radiology, Washington University School of Medicine, Saint Louis, MO, USA. [3] Computer Technologies Department, ITMO University, Saint Petersburg, Russia. [4] Université Côte d'Azur, CNRS, IPMC, Valbonne, France. [5] Université Bourgogne Franche-Comté, LNC UMR1231, Dijon, France. [6] Univ.Lille, Inserm, CHU Lille, Institut Pasteur de Lille, U1011-EGID, Lille, France. [7] Department of Internal Medicine - Nephrology, University Hospital Regensburg, Regensburg, Germany. [8] Institute of Experimental Immunology, University of Zürich, Zürich, Switzerland. [9] Department of Integrative Biology and Physiology, Center for Immunology, University of Minnesota Medical School, Minneapolis, MN, USA. [10] These authors contributed equally: Alexandre Gallerand, Marion I. Stunault. ✉email: Stoyan.ivanov@unice.fr

Monocytes are part of the mononuclear phagocytic system together with macrophages and dendritic cells (DCs)[1]. These cells are generated in the bone marrow compartment where they originate from hematopoietic stem cells (HSC) through myelopoiesis, a tightly regulated process[2]. The growth factor colony-stimulating factor 1 (CSF1), binds to its receptor CSF1R (CD115) expressed on monocytes and macrophages to promote their survival[3]. Bone marrow-derived monocytes egress to the blood circulation and this process is under the control of chemokine−chemokine receptor interactions. For instance, stromal cell-derived CXCL12 (SDF-1) binds to monocyte CXCR4 and this interaction is required for monocyte retention within the bone marrow compartment[4]. By contrast, CCL2-CCR2 interactions control monocyte egress from bone marrow to blood and their peripheral tissue recruitment from the blood circulation[5,6]. Two major subsets of blood monocytes have been described according to the expression of the cell surface marker Ly6C. Ly6C$^{high}$ monocytes, also called "classical monocytes", are recruited to peripheral tissues and can give rise to macrophages during infection, acute or chronic inflammation, or cancer and contribute to the control and the resolution of tissue inflammation[7,8]. Ly6C$^{low}$ "non-classical" monocytes patrol the blood vasculature, cleaning debris and promote healing of the injured endothelium[9]. During infection and excessive or chronic inflammation, monocytes are recruited to peripheral tissues where they differentiate into macrophages. In every organ, macrophages have been demonstrated to play a tissue-specific function. For example, splenic red pulp macrophages control iron content while testis macrophages have been implicated into spermatogenesis[10–12].

The adipose tissue has long been considered as a simple triglyceride storage organ. Various types of adipose tissue have been documented including white adipose tissue (WAT), beige adipose tissue, and brown adipose tissue (BAT). In mice, WAT is typically localized in the peritoneal cavity and subcutaneously, with its most studied depot, the perigonadal (epididymal) adipose tissue (EAT). BAT is located in the interscapular region and its main function is thermogenesis orchestrated by the mitochondrial uncoupling protein 1 (Ucp1). Beige adipose tissue is located subcutaneously (SCAT) and comprises among others the brachial and inguinal depots. Adipose tissue depot size is regulated through variations in adipocyte size and numbers. Triglycerides (TGs) are stored in lipid droplets inside adipocytes and their mobilization relies on a process named lipolysis. The first step of lipolysis is under control of the rate-limiting enzyme ATGL (adipose triglyceride lipase)[13–15] leading to TG degradation into diacylglycerol (DAG) and a fatty acid. Macrophages have been identified in BAT where they have been proposed to control neuron network density and BAT thermogenesis[16]. Nevertheless, the diversity of BAT macrophages and the mechanism responsible for their maintenance remain underappreciated. In this work, we found several populations of macrophages in BAT of wild-type mice. Additionally, we observed the existence of tissue-resident monocytes that contributed to BAT macrophage homeostasis. We used an established mouse model of adipocyte-selective Atgl-deficiency leading to BAT expansion and evaluated monocyte numbers in bone marrow and blood, and monocyte and macrophage numbers and diversity in adipose tissues. Thus, our data show that monocytes contribute to BAT expansion in a Podoplanin-dependent mechanism and favor matrix remodeling, uncovering a new function for this cell type.

## Results

### BAT macrophage diversity and monocyte contribution to their maintenance. 
To investigate the diversity of BAT myeloid cell populations, we performed single-cell RNA sequencing (scRNA-seq) analysis of cell-sorted CD45$^{+}$ cells (Fig. S1A) from C57BL/6 mice housed at room temperature and fed on a normal chow diet. These data revealed an unprecedented diversity of leukocytes residing in BAT (Fig. 1A), which can be represented by 13 unique clusters (Fig. 1A). We detected populations of T cells, B cells, and NK cells (clusters 10, 11, and 12 respectively) (Fig. 1B, C). Furthermore, among myeloid cells, we identified a population of neutrophils (cluster 0) and three separate populations of monocytes (clusters 2, 3, and 4). These subsets reflected Ly6C$^{low}$, Ly6C$^{int}$, and Ly6C$^{high}$ monocytes respectively (Fig. 1C). Of interest, we were able to identify four different macrophage populations (clusters 5, 6, 7, and 8) (Fig. 1C). These cells expressed a specific set of genes, which are represented by gene set enrichment analysis across all 13 clusters (Fig. 1C). Cluster 6 expressed many canonical alternative macrophage polarization markers including Mrc1 (CD206) and Clec10a (CD301). Additionally, this cluster was enriched in genes (C1qa, C1qb, C1qc) encoding for proteins involved in the complement pathway. Clusters 7 and 8 highly expressed genes involved in lipid metabolisms such as CD36, Lpl, and Lipa. Finally, cluster 5 was enriched in genes involved in tissue remodeling (Ecm1, MMP12, MMP19, Fn1). This BAT macrophage subset diversity might reflect their involvement in various functions from lipid handling (Clusters 6, 7, and 8) to tissue remodeling (Cluster 5). Since we detected many monocytes in our single-cell RNA sequencing data (Fig. 1A, C), we validated this observation in the BAT using CCR2$^{GFP/+}$ mice. In agreement with the transcriptomic data, we found many GFP$^{+}$ cell in BAT that were distributed among monocyte and macrophage populations (Fig. 1D). These data suggested that monocytes may contribute to BAT macrophage pool maintenance. To study this hypothesis, we used a pulse-chase model allowing to label monocytes and follow their differentiation into macrophages over a short period of time. CCR2$^{creERT2}$ mice were crossed with TdTomato reporter. To induce CCR2-driven TdTomato expression, mice were administered with tamoxifen by oral gavage and analyzed 48 h later. We observed a large fraction of TdTomato$^{+}$ cells among BAT monocytes and macrophages (Fig. 1E). Moreover, BAT Tomato$^{+}$ monocyte and macrophage counts were decreased in CCR2-deficient mice, demonstrating the relevance of this axis in BAT macrophage pool size maintenance (Fig. 1E, F). However, macrophage counts remained similar in CCR2$^{+/-}$ and CCR2$^{-/-}$ mice, suggesting that even though monocyte recruitment plays a critical role to maintain BAT macrophage pool, a compensatory mechanism, likely proliferation, could occur in the absence of monocyte recruitment to sustain constant BAT macrophage numbers (Fig. 1G).

### BAT expansion is supported by monocyte recruitment and local macrophage differentiation. 
To address how BAT monocyte and macrophage populations are affected during tissue remodeling, we generated a mouse model selectively lacking ATGL in mature adipocytes (Adipo$^{Δ/Δ}$). For this purpose, Pnpla2$^{fl/fl}$ mice were crossed with AdipoQ$^{creERT2}$ mice in order to ablate ATGL protein production specifically in mature adipocytes (Fig. S1B). To validate the efficiency of our system, those mice were crossed with TdTomato$^{fl/fl}$ reporter mice, in which the TdTomato expression reflects Cre recombinase activity (Fig. S1B). Additionally, we crossed these mice with CX3CR1$^{gfp}$ mice to facilitate analysis of monocyte distribution in tissues by flow cytometry and microscopy (Fig. S1B). Indeed, fluorescence microscopy analysis of EAT and BAT tissue sections revealed that almost all adipocytes expressed the TdTomato protein 3 weeks following tamoxifen treatment, further validating the efficiency of

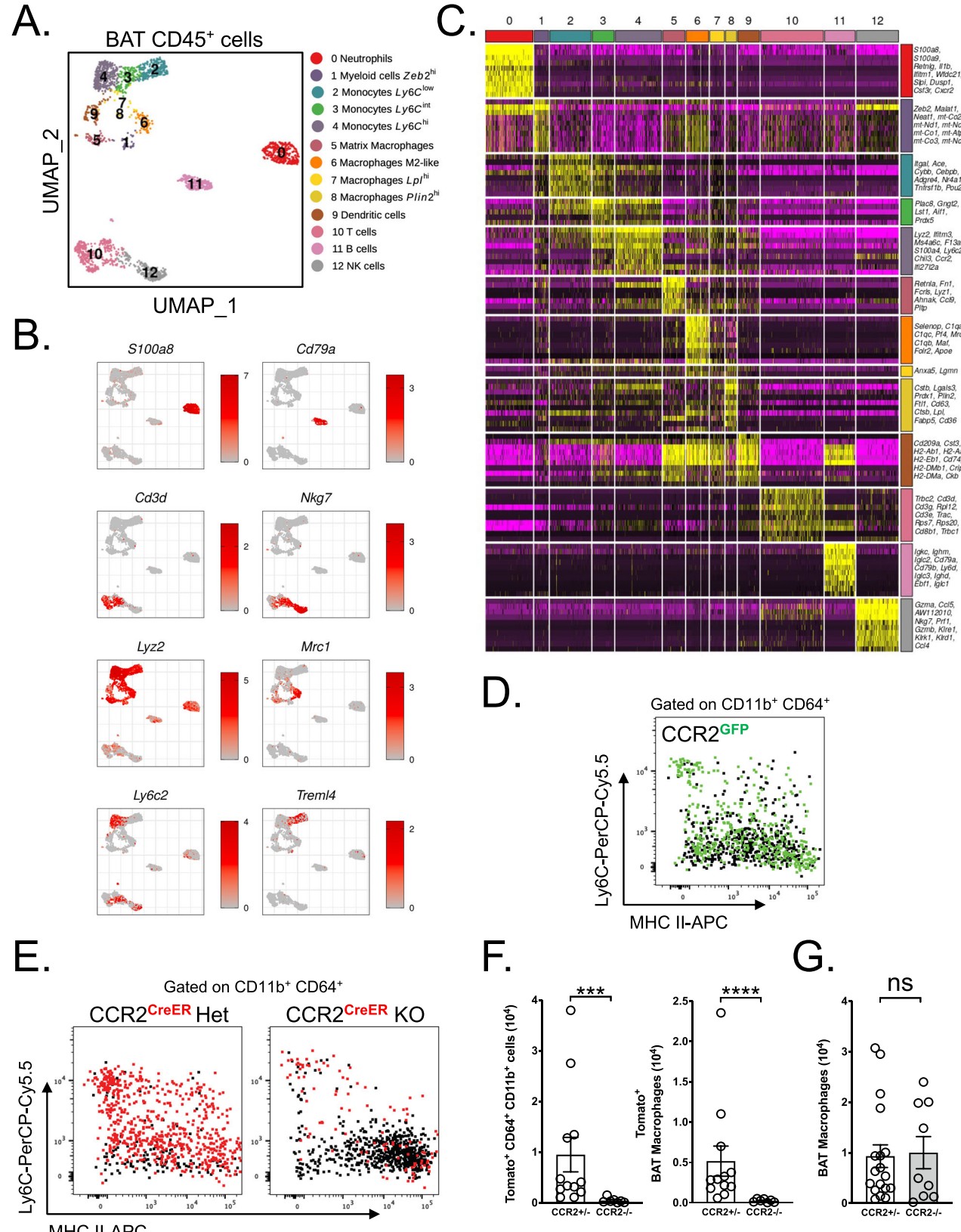

our experimental system (Fig. S1C). Western blotting analysis of BAT protein extracts showed an almost complete absence of ATGL protein in Adipo$^{\Delta/\Delta}$ mice, in comparison to littermate control animals, further confirming the deletion efficiency (Fig. S1D). The low remaining ATGL protein levels in Adipo$^{\Delta/\Delta}$ mice could originate from adipose tissue immune cells such as

macrophages or neutrophils, previously shown to express this protein[17] but in which the adiponectin promoter is not active. As expected, serum glycerol concentrations were decreased in Adipo$^{\Delta/\Delta}$ mice compared to control animals (Fig. S1E). Serum lipidomic analysis further revealed decreased concentrations of palmitic acid (C16:0), the most abundant non-esterified fatty

**Fig. 1 Monocytes contribute to BAT macrophage pool at a steady state. A** Single-cell RNA-Seq analysis of BAT CD45[+] cells from 7−8-week-old male mice. **B** UMAP representations of genes used to identify cell types among BAT CD45[+] cells. **C** Heatmap showing normalized expression levels of marker genes helping to identify cell types present in the data. **D** Flow cytometry plot showing CCR2[GFP] expression among BAT Ly6C[+/−] MHCII[+/−] cells (gated on CD45[+] CD11b[+] CD64[+] cells). **E** Flow cytometry plot showing TdTomato expression among BAT Ly6C[+/−] MHCII[+/−] cells (gated on CD45[+] CD11b[+] CD64[+] cells) from $CCR2^{creERT2/+}$ and $CCR2^{creERT2/GFP}$ mice 48 h after tamoxifen gavage. **F** Quantification of TdTomato[+] cells among CD45[+] CD64[+] CD11b[+] cells ($p = 0.0002$) and macrophages (CD45[+] CD64[+] CD11b[+] Ly6C[−]) ($p < 0.0001$) in the BAT of $CCR2^{creERT2/+}$ ($CCR2^{+/−}$, $n = 12$) and $CCR2^{creERT2/GFP}$ ($CCR2^{−/−}$, $n = 7$) mice 48 h after tamoxifen gavage. **G** Quantification of macrophages in the BAT of $CCR2^{+/−}$ ($n = 18$) and $CCR2^{−/−}$ ($n = 9$) mice, $p = 0.8997$. Panels **D**, **E**, **F**, and **G** represent pooled data from two independent experiments. All data are represented in means ± SEM. Two-tailed Mann−Whitney tests were used to determine statistical significance. ns $p > 0.05$; *$p < 0.05$; **$p < 0.01$. Source data are provided as a Source Data file.

acids (NEFA) (approximatively 33% of total NEFAs). Myristic (C14:0), and α-linolenic (C18:3 n-3) acids were also decreased (Fig. S1F). Interestingly, other types of fatty acids were not modulated in Adipo[Δ/Δ] mice when compared to littermate control animals (Fig. S1G).

In agreement with previous reports using constitutive *Atgl* deletion[18,19], we found a strikingly increased BAT weight (×2.5 fold) in Adipo[Δ/Δ] mice in comparison to co-housed littermate controls after 3 weeks of Cre induction (Fig. 2A). Nevertheless, EAT had a similar weight in both genotypes (Fig. 2A). Tissue histology analysis revealed widespread changes in brown adipocyte morphology with the appearance of larger lipid droplets, a process named "whitening" of the brown adipose depot (Fig. 2B)[20]. We also observed the appearance of CD11b[+] cells in the BAT of Adipo[Δ/Δ] mice, in patterns similar to "crown-like structures" reported in EAT during obesity (Fig. 2C)[20]. These cells were rarely detected in control mice (Fig. 2C).

To further define whether specific immune cell subsets were enriched in Adipo[Δ/Δ] mice, we performed a comparative single-cell RNA-seq analysis between cell-sorted CD45[+] cells from control and adipocyte-*Atgl*-deficient mice. We noticed a major and specific enrichment in all macrophage clusters in Adipo[Δ/Δ] animals without affecting the populations of B cells, T cells, and NK cells (Fig. 2D, E). This observation indicated that BAT expansion induced a macrophage-specific enrichment in the organ. Moreover, a distinctive fatty acid-centered metabolic signature was detected in these macrophage subsets (Figs. 2F and S2A). Using the Slingshot tool, we established a predictive differentiation model linking monocytes to macrophage clusters (Fig. 2G). This model suggested that Ly6C[high] monocytes give rise to two intermediate macrophage populations (clusters 7 and 8). Two terminally differentiated macrophage clusters (clusters 5 and 6) were identified and corresponded to clusters involved in lipid metabolism and matrix remodeling (Fig. 2G). In line with our pathway enrichment analysis, these cells displayed higher expression of genes involved in lipid handling including *Plin2*, *CD36*, *Trem2*, and *Lpl* in Adipo[Δ/Δ] mice compared to controls (Fig. 2H). Although *Mrc1* expression was found on all macrophage subsets, higher expression of this marker was observed on cluster 6 (Fig. S2B). On the other hand, cluster 5 macrophages were found to specifically express *CD226* (Fig. S2B).

To phenotypically confirm the data from our scRNAseq analysis, we performed a flow cytometry analysis of adipose tissues obtained from control and Adipo[Δ/Δ] mice. Since tissue expansion did not occur in the EAT of Adipo[Δ/Δ] mice, we investigated macrophage and monocyte phenotypes in both the BAT and EAT of control and Adipo[Δ/Δ] mice. We applied an established gating strategy to separately analyze macrophages and monocytes (Fig. S2C)[21,22]. Previous reports demonstrated the presence, at a low frequency, of monocytes residing in tissues[22]. In our study, we identified monocytes as CD45[+]CD64[+]MerTK[−]CD11b[+] cells and macrophages as CD45[+]CD64[+]MerTK[+] (Fig. S2C). Furthermore, we developed a gating strategy for the in-vivo identification of cluster 5 and cluster 6 cells, namely CD226[high] and CD206[high] macrophages (Fig. S2D).

The proportion of CD206[high] macrophages was equivalent in BAT and EAT from Adipo[Δ/Δ] mice. However, we found that the population of CD226[high] macrophages was markedly enriched in BAT compared to EAT (Fig. S2D). Increased monocyte and macrophage counts were observed in the BAT of Adipo[Δ/Δ] mice (Fig. 2I), but not in their EAT (Fig. S2E). To apprehend whether this accumulation was triggered solely by the loss of BAT lipolysis or could be linked to ATGL deficiency in other adipose depots, we analyzed BAT myeloid cells in $Ucp1^{cre} \times Pnpla2^{fl/fl}$ (BAT[Δ/Δ]) mice. *Ucp1* is highly expressed in brown adipocytes and lacking in white adipocytes. BAT monocytes and macrophages were more numerous in BAT[Δ/Δ] mice in comparison to controls, suggesting that the Adipo[Δ/Δ] BAT phenotype was driven by BAT-mediated cues (Fig. S2F). Moreover, BAT monocytes in Adipo[Δ/Δ] mice expressed higher levels of MHC II, a marker that is upregulated in monocytes that recently egressed the blood vasculature and entered an organ (Fig. 2J)[22]. However, their CD11c expression remained similar between both genotypes (Fig. 2J). Importantly, MHC II expression was comparable between EAT monocytes obtained from control and Adipo[Δ/Δ] mice (Fig. S2G). As a whole, BAT macrophages from Adipo[Δ/Δ] mice, but not EAT macrophages, also displayed altered expression of the canonical alternative polarization markers CD206 and CD301 (Figs. 2K and S2H). This may be representative of the increased presence of CD206[low] CD301[low] macrophages (mainly in clusters 7 and 8), as *Mrc1* and *Clec10a* expression did not appear to be diminished in M2-like macrophages (cluster 6) (Fig. S2I). The expression of CD11c, a classical M1 activation marker, was not altered (Fig. 2K). Taken together, this demonstrated that ATGL-deletion in adipocytes triggers monocyte BAT recruitment and leads to increased numbers of BAT macrophages with a specific phenotype.

Local proliferation of tissue-resident macrophages is involved in their pool maintenance[23]. To investigate macrophage proliferation rate, we performed Ki67 staining and found a similar level of proliferation in BAT macrophages from control and Adipo[Δ/Δ] mice (Fig. 2L). This observation was also supported by the analysis of our single-cell RNA sequencing data that revealed that few BAT macrophages show expression of proliferation-related genes including *Mki67*, *Ccna2,* and *Top2a* (Fig. S2J).

**BAT-selective monocyte recruitment leads to diminished blood monocyte counts without affecting bone marrow hematopoiesis**. We next asked whether increased monocyte counts in Adipo[Δ/Δ] mice are specific to BAT or could also be observed in other peripheral tissues. To monitor monocyte distribution across tissues we took advantage of a CCR2-targeting radiotracer [64]Cu-DOTA-ECL1i for non-invasive positron emission tomography (PET)/computed tomography (CT) imaging of CCR2[+] cells distribution across multiple tissues without the biases of tissue digestion. This provides the opportunity to simultaneously investigate numerous organs, thus limiting the risk to miss an unexpected tracer accumulation in a specific location[24,25]. Representative PET/CT images revealed significantly increased signal, indicative of CCR2 accumulation, in the BAT of Adipo[Δ/Δ]

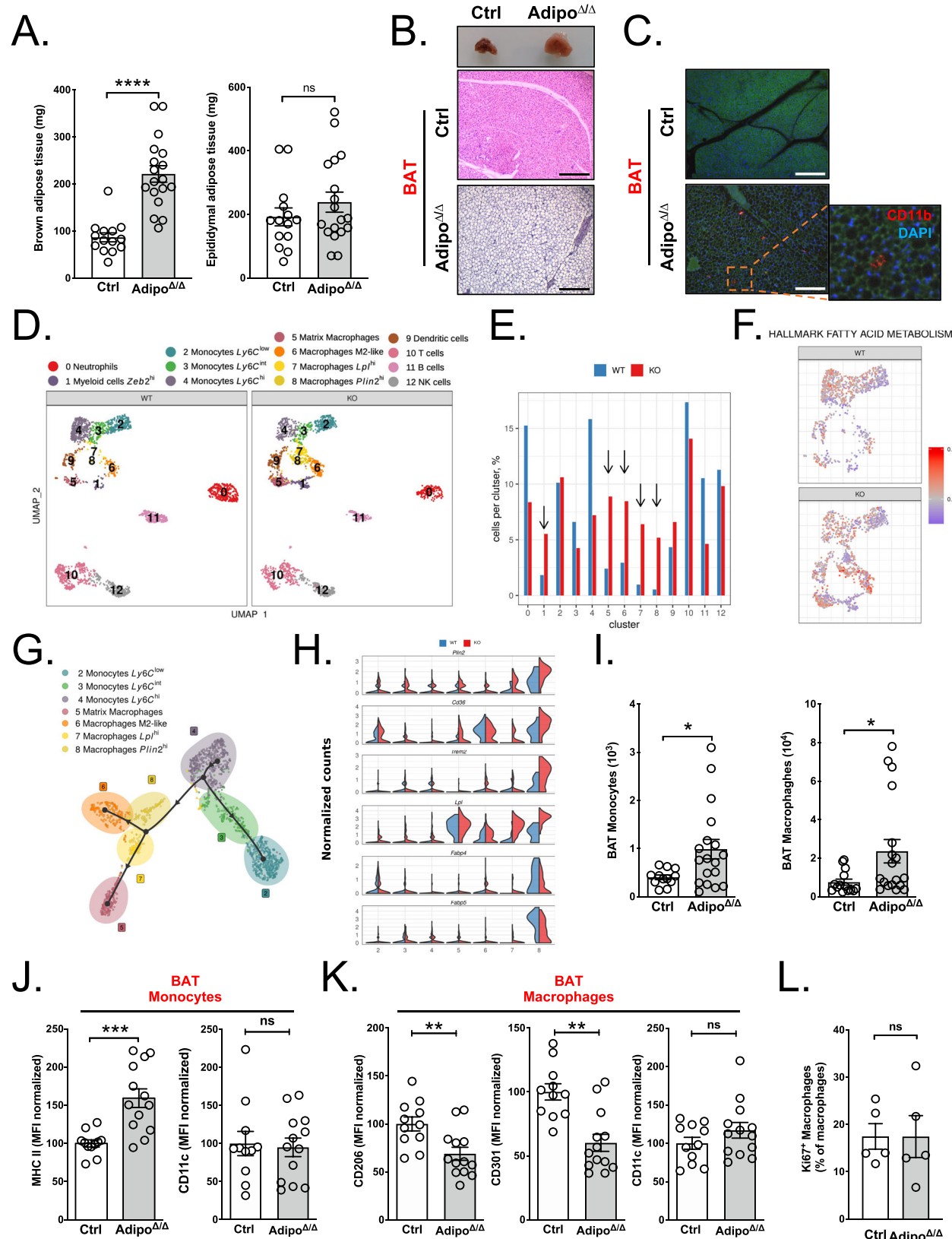

mice at 16 days, but not at 2 days, post-tamoxifen administration (Cre⁺TAM⁺) (Fig. 3A, B). These data confirm the progressive monocyte infiltration in BAT after Atgl-deletion. Of note, we observed a comparable level of BAT CCR2 signal in both Adipo$^{Cre+}$ animals treated with vehicle (oil) (Cre⁺TAM⁻) and Adipo$^{+/+}$ mice treated with tamoxifen (Cre⁻TAM⁺) (Fig. 3A, B).

To analyze monocyte density, the biodistribution of the tracer was measured across tissues and we observed that CCR2 signal was selectively increased in the BAT of Adipo$^{Δ/Δ}$ mice (Fig. 3C). Despite a detectable signal in several organs, we did not observe a differential CCR2 distribution in any other tissue (Fig. 3C). Of

**Fig. 2 Monocytes and macrophages accumulate in the BAT during tissue expansion. A** BAT and EAT weight in control ($n = 14$) and Adipo$^{\Delta/\Delta}$ ($n = 18$) mice. $p < 0.0001$ (left) and $p = 0.4582$ (right). **B** Representative images of BAT morphology and analysis using H&E staining in Adipo$^{\Delta/\Delta}$ mice and controls. **C** CD11b staining (red) in the BAT of Adipo$^{\Delta/\Delta}$ and control mice. The presence of CLS is highlighted in the magnified box. **D** Single-cell RNA-Seq analysis of BAT CD45$^+$ cells from 7−8-week-old control and Adipo$^{\Delta/\Delta}$ male mice. **E** Proportion of each cluster identified in scRNA-Seq analysis. **F** Single-cell RNA-Seq analysis of the "fatty acid metabolism" Hallmark gene set expression. **G** Differentiation model of BAT myeloid cells generated using the Slingshot tool. **H** Violin plots showing RNA-Seq analysis of *Plin2*, *CD36*, *Trem2*, *Lpl*, *Fabp4*, and *Fabp5* expression by BAT macrophages and monocytes from Adipo$^{\Delta/\Delta}$ and control mice. **I** Quantification of BAT monocyte and macrophage numbers in control ($n = 12$) and Adipo$^{\Delta/\Delta}$ ($n = 18$) mice using flow cytometry. $p = 0.0346$ (left) and $p = 0.0143$ (right). **J** Quantification of surface MHC II and CD11c expression by BAT monocytes in control ($n = 11$) and Adipo$^{\Delta/\Delta}$ ($n = 13$) mice using flow cytometry. $p = 0.0004$ (left) and $p = 0.8646$ (right). **K** Quantification of surface CD206, CD301, and CD11c expression by BAT macrophages in control ($n = 11$) and Adipo$^{\Delta/\Delta}$ ($n = 13$) mice using flow cytometry. $p = 0.0031$ (left), $p = 0.0012$ (middle), $p = 0.3607$ (right). **L** Proportion of Ki-67-expressing macrophages in the BAT of control ($n = 5$) and Adipo$^{\Delta/\Delta}$ ($n = 5$) mice analyzed by flow cytometry. $p = 0,6905$. Panels **A**, **I**, **J**, and **K** represent pooled data from four independent experiments. Panels **B**, **C** are representative of two independent experiments. Panel **L** represents data from one experiment. All data are represented in means ± SEM. Two-tailed Mann−Whitney tests were used to determine statistical significance. ns $p > 0.05$; *$p < 0.05$; **$p < 0.01$; ***$p < 0.001$; ****$p < 0.0001$. Source data are provided as a Source Data file.

interest, blood CCR2 signal was reduced, even though the difference was not significant, in Adipo$^{\Delta/\Delta}$ animals (Fig. 3C). Ex vivo tissue collections showed an enlarged mass of BAT in Adipo$^{\Delta/\Delta}$ animals compared to the other two groups. Moreover, autoradiography showed specific tracer uptake in the BAT and not in the surrounding white adipose tissue, thus supporting PET data (Fig. S3A). Taken together, this suggests that adipocyte-specific Atgl deletion leads to BAT-selective monocyte recruitment. Thus, disruption of BAT homeostasis leads to a very specific monocyte recruitment without affecting any other peripheral organ.

Because we observed a tendency to decreased CCR2 signal in blood (Fig. 3C), we next wondered whether BAT monocyte recruitment could affect the pool of circulating blood monocytes. Using flow cytometry analysis, we detected a marked decrease in the numbers of blood monocytes (CD45$^+$CD115$^+$CD11b$^+$ cells) (Fig. 3D and Fig. S3B). This observation was valid for both Ly6C$^{hi}$ and Ly6C$^{lo}$ monocyte subsets (Fig. 3E). However, neutrophil (CD45$^+$Gr1$^+$CD11b$^+$CD115$^{low}$), B cell (CD45$^+$CD19$^+$) and T cell (CD45$^+$TCRβ$^+$) numbers were similar in ATGL-sufficient and deficient mice (Fig. S3C). Likewise, CCL2 and CXCL12 serum levels were not modulated by the loss of adipose tissue lipolysis, excluding systemic inflammation as a cause for the decrease in blood monocyte numbers (Fig. 3F). We thus hypothesized that reduced blood monocyte numbers in ATGL-deficient mice result from either an altered hematopoiesis or mature monocyte export from bone marrow to blood. Flow cytometry analysis revealed no substantial difference in the number of monocyte precursor cells including LSK (Lin$^-$Sca1$^+$cKit$^+$) (Fig. S3D). Bone marrow monocyte counts were also very similar in control and adipocyte ATGL-deficient mice (Fig. S3D). Thus, blood monocyte number reduction is unlikely to result from defects in bone marrow hematopoiesis or stromal niche alteration. Since the pool of bone marrow monocytes results from the dynamic interaction between monocyte proliferation, death, and export to peripheral blood, we analyzed these parameters. Monocyte proliferation was comparable in Adipo$^{\Delta/\Delta}$ and littermate control mice (Fig. S3E). Annexin V staining on bone marrow monocytes suggested a similar level of apoptosis in both genotypes, ruling out a role for cell death (Fig. S3E). To analyze monocyte export from bone marrow to blood, we i.v. injected mice with BrdU and analyzed the percentage of BrdU positive monocytes in blood 16 h postinjection. We found that approximately 70% of blood monocytes had undergone proliferation by incorporating BrdU (Fig. S3F) in both control and Adipo$^{\Delta/\Delta}$ mice. Therefore, we concluded that the monocyte export rate is not modified during BAT expansion and could not account for diminished blood monocyte numbers. To apprehend the mechanisms that could account for increased brown adipose tissue monocyte numbers, we performed real-time PCR analysis. We found an increased expression of the

pro-inflammatory gene *tnfa* and the chemokine *ccl2* in the BAT of Adipo$^{\Delta/\Delta}$ mice (Fig. 3G). The latest is responsible for monocyte recruitment to tissues and could explain increased numbers of BAT monocytes and macrophages. Although ELISA analysis of BAT homogenates failed to confirm an increase in TNFα levels, CCL2 protein levels were markedly augmented in the BAT of Adipo$^{\Delta/\Delta}$ mice (Fig. 3H). Together, these data suggest the onset of a very low-grade tissue inflammation with specific CCL2-mediated monocyte recruitment to the BAT of Adipo$^{\Delta/\Delta}$ mice.

**Monocyte depletion prevents BAT expansion and sustains tissue browning.** To address whether BAT monocyte recruitment is involved in tissue remodeling and growth, we depleted blood monocytes via administration of anti-CCR2 antibody (MC-21)[26] (Fig. 4A). Mice received tamoxifen treatment and were then allowed to rest for one week during which BAT started expanding. MC-21 or vehicle treatment were then administered daily for 5 days. Blood and tissues were analyzed 16 h after the last MC-21 injection (Fig. 4A). This procedure led to the complete depletion of blood monocytes as previously described[26] (Fig. S4A). The spleen monocyte population was also completely depleted in MC-21-treated mice (Fig. S4A). Flow cytometry analysis demonstrated a major monocyte depletion in the BAT of MC-21-treated Adipo$^{\Delta/\Delta}$ animals in comparison to vehicle-administered controls (Fig. 4B). BAT macrophage numbers were slightly, but not significantly decreased in MC-21-treated Adipo$^{\Delta/\Delta}$ mice (Fig. 4C). Importantly, BAT weight was significantly decreased, by approximatively 40%, in Adipo$^{\Delta/\Delta}$ mice treated with anti-CCR2 Ab in comparison to vehicle administered Adipo$^{\Delta/\Delta}$ animals (Fig. 4D). We observed that in control mice, in which BAT is not expanding, CCR2 depletion had no effect on tissue weight (Fig. 4D). EAT weight remained similar among MC-21-treated and vehicle-treated animals (Fig. S4B). Tissue histology analysis revealed that EAT adipocyte size was similar in control and Adipo$^{\Delta/\Delta}$ mice (Fig. S4B). BAT histology analysis revealed that MC-21 administration exacerbated the appearance of CLS (Fig. S4C). MC-21 treatment had no impact on serum glycerol, TG and NEFA levels (Fig. S4D). However, we detected numerous multilocular zones in MC-21-treated animals in comparison to vehicle-treated Adipo$^{\Delta/\Delta}$ mice (Fig. 4E), suggesting that monocyte recruitment to BAT modulates tissue morphology and favors BAT "whitening". Whether this mechanism is solely responsible for BAT weight decrease in MC-21-treated mice remains to be established.

**Monocytes regulate BAT expansion via Podoplanin engagement.** Previous data demonstrated that myeloid cells could modulate tissue expansion by interacting with stromal fibroblastic

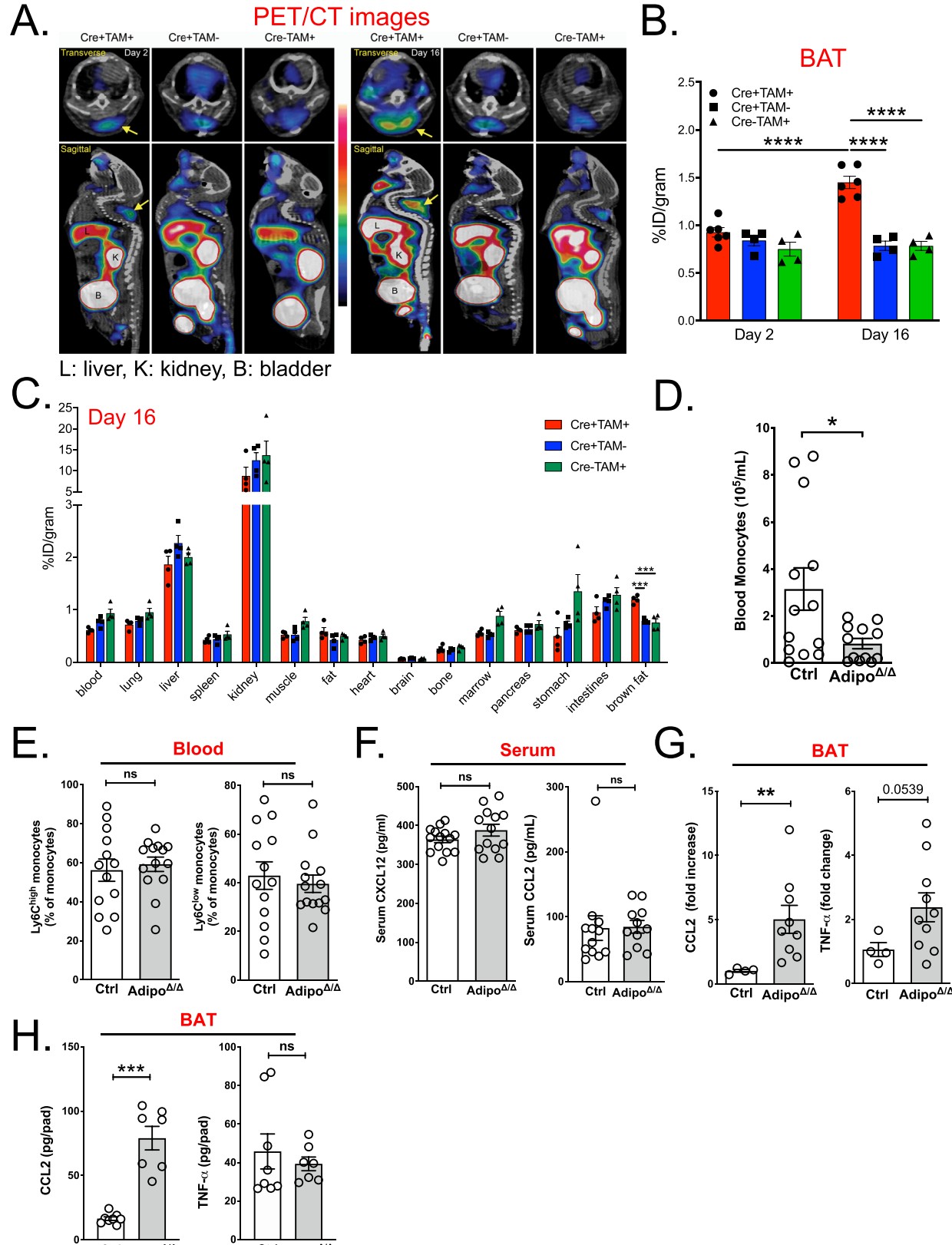

L: liver, K: kidney, B: bladder

reticular cells (FRCs)[27]. Thus, we sought to determine whether monocytes in BAT might interact with fibroblasts to favor tissue expansion. Using α-SMA staining, we identified the presence of α-SMA+ cells with a fibroblastic morphology in the BAT of Adipo$^{\Delta/\Delta}$ mice, suggesting the existence of activated fibroblasts in expanding BAT (Fig. S5A). To test whether monocytes could interact with fibroblasts, we co-cultured mouse embryonic fibroblasts (MEFs) with primary blood monocytes. Monocyte addition induced an increase in fibroblast cellular area (Fig. 5A, B). Importantly, we observed increased numbers of cell protrusions in MEFs cultured in the presence of blood monocytes (Fig. 5A, B). We next investigated whether this interaction

**Fig. 3 Adipocyte ATGL deletion induces a specific recruitment of blood monocytes to the BAT. A** $^{64}$Cu-DOTA-ECL1i PET/CT images analyzing tracer uptake in control and Adipo$^{\Delta/\Delta}$ mice 2 days (left panel) and 16 days (right panel) post-tamoxifen administration ($n = 4$–$6$/group). L: liver; B: bladder; K: kidney. **B** Quantification of tracer uptake in the BAT of Cre$^-$ tamoxifen-treated (Cre$-$ Tam+, $n = 4$) and Cre$^+$ vehicle-treated (Cre+ Tam$-$, $n = 4$) controls and Adipo$^{\Delta/\Delta}$ animals (Cre+ Tam+, $n = 6$) 2- and 16-days post-tamoxifen administration. $p < 0.0001$ for each comparison. **C** Biodistribution of $^{64}$Cu-DOTA-ECL1i in Cre$-$ Tam+ and Cre+ Tam-controls and Adipo$^{\Delta/\Delta}$ animals 16 days post-tamoxifen administration (n = 4/group). $p < 0,001$ for each comparison in BAT. **D** Quantification of blood monocyte counts in control ($n = 13$) and Adipo$^{\Delta/\Delta}$ ($n = 12$) mice using flow cytometry. $p = 0.0384$. **E** Proportions of Ly6C$^{high}$ and Ly6C$^{low}$ monocytes among total blood monocytes in control ($n = 13$) and Adipo$^{\Delta/\Delta}$ ($n = 14$) mice using flow cytometry. $p = 0.6940$ (left) and $p = 0.6160$ (right). **F** Analysis of serum CCL2 and CXCL12 in control ($n = 14$) and Adipo$^{\Delta/\Delta}$ ($n = 13$) mice by ELISA. $p = 0.2388$ (left) and $p = 0.3793$ (right). **G** Analysis of BAT *CCL2* and *TNFα* expression in control ($n = 4$) and Adipo$^{\Delta/\Delta}$ ($n = 9$ and 10 respectively) mice by qPCR. $p = 0.0028$ (left) and $p = 0.0539$ (right). **H** Quantification of CCL2 and TNFa protein levels in BAT homogenates from control ($n = 8$) and Adipo$^{\Delta/\Delta}$ ($n = 7$) mice by ELISA. $p = 0.0003$ (left) and $p = 0.6126$ (right). Panels **D**, **E** represent pooled data from four independent experiments. Panels **F**, **G** represent pooled data from two independent experiments. Panel **H** represents data from one experiment. All data are represented in means ± SEM. Ordinary one-way ANOVA with Bonferroni post-test were used to determine statistical significance in panels **B**, **C**. Two-tailed Mann−Whitney tests were used to determine statistical significance in panels **D**−**H**. ns $p > 0.05$; *$p < 0.05$; **$p < 0.01$; ***$p < 0.001$; ****$p < 0.0001$. Source data are provided as a Source Data file.

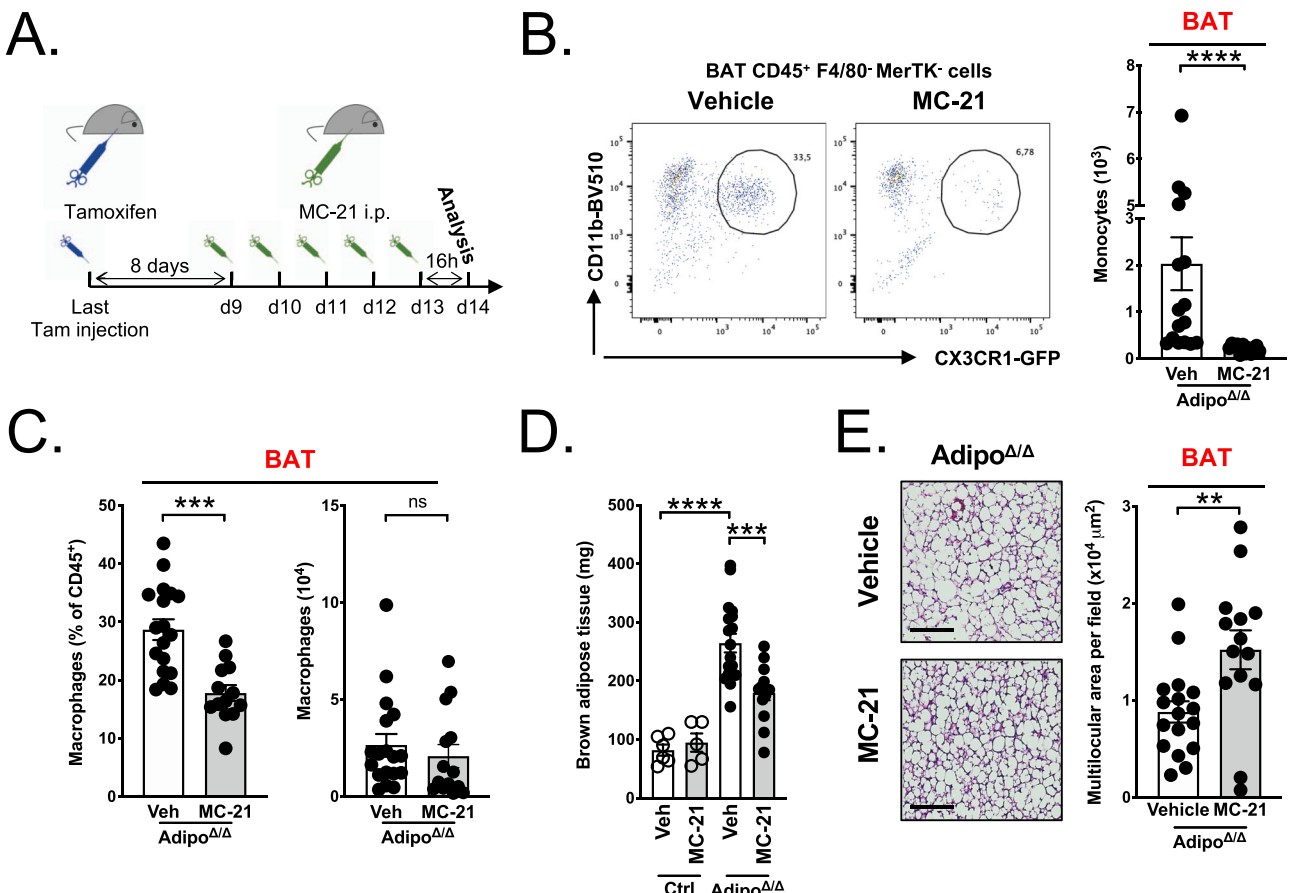

**Fig. 4 Monocyte depletion prevents the expansion of lipolysis-deficient BAT. A** Schematic representation of the experimental procedure used for MC-21-mediated monocyte depletion. **B** Representative dot plots (left) and quantification (right) of BAT monocytes in MC-21 ($n = 12$) or vehicle-treated ($n = 16$) Adipo$^{\Delta/\Delta}$ mice. $p < 0.0001$. **C** Frequency and numbers of BAT macrophages in MC-21 ($n = 13$) or vehicle-treated ($n = 18$) Adipo$^{\Delta/\Delta}$ mice. $p = 0.0001$ (left) and $p = 0.3189$ (right). **D** Brown adipose tissue weight in MC-21 ($n = 5$) or vehicle-treated ($n = 6$) control mice, and MC-21 ($n = 14$) or vehicle-treated ($n = 18$) Adipo$^{\Delta/\Delta}$ mice. $p < 0.0001$ (vehicle-treated control and Adipo$^{\Delta/\Delta}$ mice) and $p = 0,0002$ (vehicle-treated and MC-21-treated Adipo$^{\Delta/\Delta}$ mice). **E** Representative images of H&E-stained BAT and quantification of multilocular areas in MC-21 ($n = 14$) or vehicle-treated ($n = 17$) Adipo$^{\Delta/\Delta}$ mice. Scale bar = 100 μm. $p = 0.0053$. Data were derived from three pooled independent experiments. All data are represented in means ± SEM. Two-tailed Mann−Whitney tests were used to determine statistical significance. ns $p > 0.05$; *$p < 0.05$; **$p < 0,01$; ***$p < 0.001$. Source data are provided as a Source Data file.

required a cell-to-cell contact, or alternatively was mediated by soluble mediators. In a transwell experiment, allowing for physical separation between fibroblasts and monocytes, we observed that fibroblast spreading occurred even in the absence of direct contact with monocytes (Fig. 5A, B). These results indicated that

monocyte interaction with fibroblasts induced changes in their morphology. Therefore, we investigated whether these morphological changes were paralleled by modulations of fibroblast mechanical properties. Using a traction force microscopy approach, we observed that fibroblasts co-cultured with

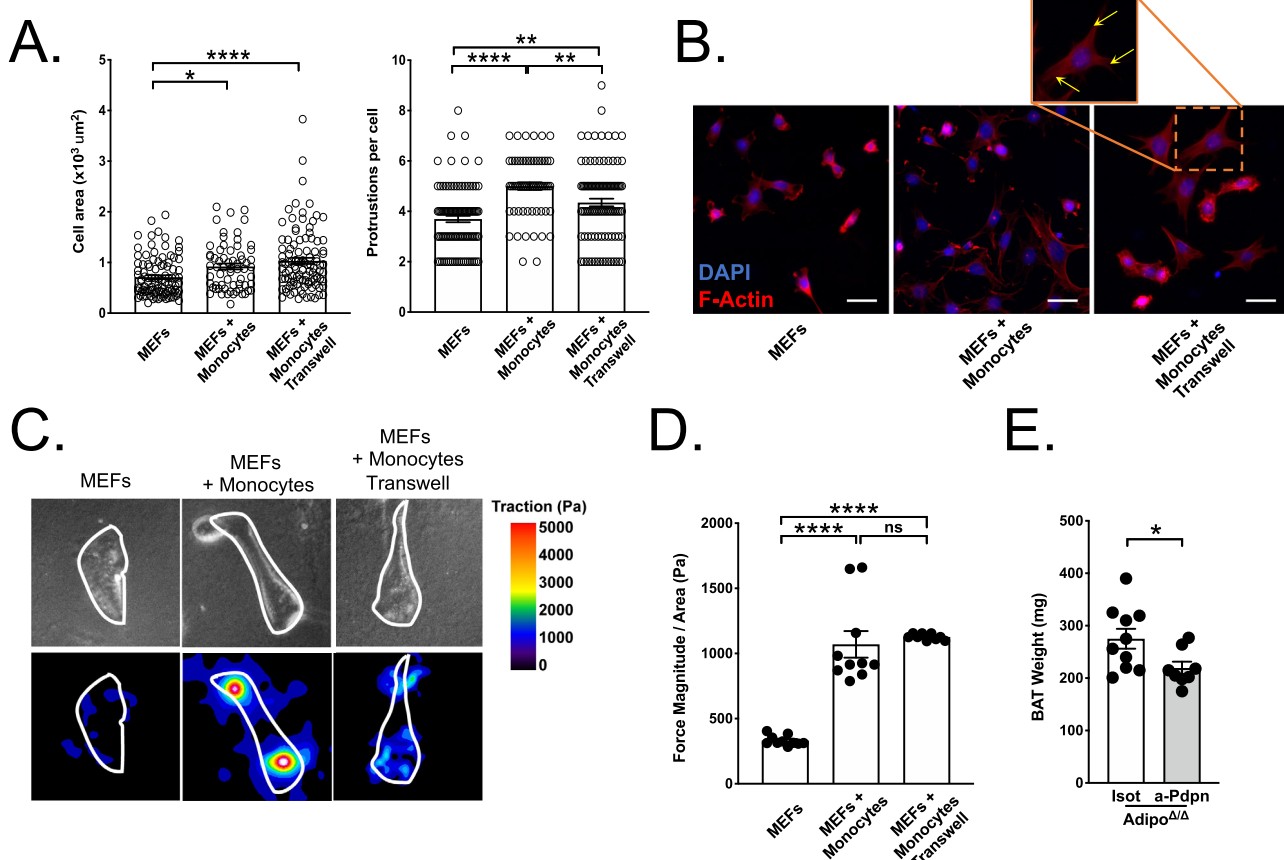

**Fig. 5 Monocyte interaction with fibroblasts through the Podoplanin axis is required for BAT expansion. A** Quantification of MEF morphological features after a 18 h co-culture experiment with monocytes placed in the same well or in a transwell insert with 0.4 μm pores. Each dot represents a cell. $n = 101$ (MEFs), 64 (MEFs + Monocytes) and 103 (MEFs + Monocytes Transwell) cells representative of three independent experiments. *: $p = 0.0257$, ****: $p = $ <0.0001 (left). **: $p < 0.01$, ****: $p < 0.0001$(right). **B** Representative images of MEF morphology after a 18 h co-culture experiment with monocytes placed in the same well or in a transwell insert with 0.4 μm pores. F-actin was revealed using phalloidin staining. Yellow arrows indicate protrusions. Scale bar = 20 μm. Representative heat map (**C**) and quantification (**D**) showing contractile forces generate by MEFs plated on 8 kPa hydrogel after a 18 h co-culture experiment with monocytes placed in the same well or in a transwell insert with 0.4 μm pores. Mean of $n = 10$ wells per condition. ****: $p < 0.0001$. **E** Brown adipose tissue weight in anti-Podoplanin ($n = 8$) or isotype control-treated ($n = 10$) Adipo$^{\Delta/\Delta}$ mice. $p = 0.0363$. Panels **A**, **B** are representative of three independent experiments. Panel **D** is representative of two independent experiments. Panel **E** represents pooled data from two independent experiments. All data are represented in means ± SEM. Ordinary one-way ANOVA with Bonferroni post-test were used to determine statistical significance in panels (**A**, **D**). A two-tailed Mann−Whitney test was used to determine statistical significance in panel (**E**). ns $p > 0.05$; *$p < 0.05$; **$p < 0.01$; ***$p < 0.001$. Source data are provided as a Source Data file.

monocytes exerted increased traction forces on their substrate (Fig. 5C, D). This observation was also repeated when using a transwell system, confirming that cell-contact between MEFs and monocytes was not required for modulations of fibroblast activation (Fig. 5C, D).

Myeloid cells were previously shown to favor tissue expansion through the CLEC-2-Podoplanin (Pdpn) axis[27]. CLEC-2 expression was documented on immune cells, including dendritic cells[28,29]. We aimed to investigate whether BAT immune cells express CLEC-2. Flow cytometry analysis revealed that BAT macrophages, and to a lesser degree BAT monocytes, express detectable CLEC-2 levels on their cell-surface (Fig. S5B). We next decided to inquire whether Pdpn engagement is involved in BAT expansion. When we included a Pdpn-blocking antibody to our in vitro co-culture system, we observed that fibroblast spreading was prevented, demonstrating the role of these factors in our system (Fig. S5C). In vivo, Pdpn blocking blunted BAT expansion in Adipo$^{\Delta/\Delta}$ mice, similarly to MC-21 treatment (Fig. 5E). Flow cytometry analysis demonstrated that administration of Pdpn blocking Ab had no effect on monocyte and macrophage

numbers in BAT (Fig. S5D). This observation was not surprising since Pdpn engagement occurs downstream of monocyte BAT recruitment. Furthermore, we analyzed serum NEFA, glycerol, and TG levels in Adipo$^{\Delta/\Delta}$ animals following Pdpn blockade or control treatment. We detected a similar concentration of these metabolites suggesting that blocking Pdpn engagement prevents BAT expansion without affecting systemic NEFA, glycerol, and TG levels (Fig. S5F). Taken together, this set of data identified the Pdpn axis as a central regulator of BAT expansion.

## Discussion
Although macrophages were detected in previous single-nuclei RNA-seq analyses of murine and human whole BAT[30,31], their proportions among total BAT cells were too small to gain detailed insights into their diversity. Using scRNA-seq analysis of BAT CD45$^+$ cells, the present study uncovered the co-existence of several macrophage and monocyte subsets in healthy BAT. We found that monocytes intensely contribute to BAT macrophage maintenance. In a genetic model of BAT expansion, we observed

a massive monocyte and macrophage recruitment that sustained changes in tissue morphology. This myeloid cell recruitment was BAT-specific and did not affect other adipose tissue depots (EAT) or peripheral tissues. Tissue inflammation remained low as TNFα levels were unchanged in the expanding BAT. However, CCL2 accumulation was paralleled by increased CCR2 signal, suggesting a key role for the CCL2-CCR2 axis in BAT monocyte recruitment. CX3CR1−CX3CL1 interactions have been proposed to be involved in WAT homeostasis. Whether this axis plays a role in BAT macrophage homeostasis remains to be defined. We did not observe a systemic inflammation but found a surprising decrease in blood monocyte counts. Importantly, in this scenario medullary hematopoiesis was not affected. Monocyte depletion compromised BAT expansion and modulated BAT architecture, thus suggesting a key role of monocytes or monocyte-derived macrophages during BAT expansion. Whether a specific population of BAT monocytes or macrophages is responsible for BAT expansion remains to be defined.

Monocytes play a key role, beneficial or detrimental, during many pathological conditions, and controlling their numbers and functions could improve disease outcome. For example, during cardiovascular disease development, monocyte counts are an independent risk factor[32,33]. Mice living at thermoneutrality (30 °C) have lower peripheral blood monocytes[25]. Interestingly, in this scenario monocytes accumulated inside the bone marrow. This was paralleled by a lower CCR2 signal in brown fat, suggesting altered monocyte recruitment in this organ[25]. Importantly, the recruitment of CCR2+ cells to subcutaneous adipose tissue, but not to BAT contributes to adaptive thermogenesis during cold exposure[34]. Whether monocyte recruitment to BAT favors thermogenesis and heat dissipation remains to be established. In our study, we revealed that monocyte depletion favors a multilocular phenotype in BAT adipocytes. This parameter might reflect local thermogenesis. In a mouse model of burn injury, CCR2+ cells were also shown to contribute to adipose tissue thermogenesis[35]. On the other hand, during cold exposure blood monocyte counts are increased in both mice and humans and this is mirrored by augmented adipose tissue lipolysis[25]. Mice with constitutive AdipoQ-driven ATGL deficiency display reduced BAT Ucp1 expression[36], paralleled by increased CCL2 and TNFα expression[37]. We did not detect significant BAT inflammation in our short-term tamoxifen-induced setup, but rather a very specific CCL2-mediated monocyte recruitment. The precise cellular source of BAT-produced CCL2, and its contribution to tissue architecture remain to be established. Brown fat lipolysis is dispensable for thermogenesis, and Ucp1 levels remain stable in mice specifically lacking BAT lipolysis[18,19]. Importantly, accumulation of BAT macrophages and monocytes still occurred in BAT^Δ/Δ animals, suggesting that local, rather than systemic ATGL-deletion, leads to monocyte recruitment.

Monocyte recruitment to BAT allows optimal tissue expendability. Previous work established the involvement of myeloid cells in the control of tissue contractility and the ability to expand[27,38]. The engagement of the Podoplanin-CLEC-2 axis modulates lymph node fibroblastic reticular cells stiffness and favors lymph node relaxation. Whether this mechanism occurs in other tissues, and in BAT, in particular, remains to be defined. We were able to detect CLEC-2 expression on BAT macrophages and monocytes. Moreover, we observed that blocking podoplanin prevented BAT expansion in vivo. Extracellular matrix modulations during diabetes modulate key adipocyte metabolic functions including lipolysis and glucose handling[39]. One could expect that macrophages, and notably CD226high macrophages from cluster 5, are involved in this process and could regulate BAT function. Establishing genetic or pharmacological models allowing for the selective depletion of these cells will provide a unique opportunity to define their precise function.

BAT expansion and whitening are characterized by CLS appearance and collagen accumulation[20]. Whether monocytes participate to BAT CLS formation or, alternatively, locally resident BAT macrophages are solely involved in this mechanism, remains to be completely established. Our data show that depletion of CCR2+ cells, likely monocytes, leads to increased frequency of CLS in the BAT of Adipo^Δ/Δ mice. This might be due to the accumulation of dead or dying adipocytes in the absence of monocyte BAT recruitment, suggesting that monocytes play a key role in maintaining BAT homeostasis. Limited BAT expansion in CCR2-depleted mice could also be due to affected adipocyte lineage generation. Whether adipocyte generation is altered in monocyte-depleted animals is an intriguing question that is beyond the scope of interest of the current manuscript.

Thus, controlling macrophage numbers in healthy and inflamed tissues seems critical. Previous work demonstrated the existence of brown adipose tissue-resident macrophages. These cells have been shown to control local neuron network density and the subsequent norepinephrine production and thermogenesis[16]. We found here that healthy and inflamed BAT contains four different macrophage subsets. Further work is required to establish the precise function of each population through the generation of specific genetic models or pharmacological inhibitors.

## Methods

### Mice

*Generation of Adipo^Δ/Δ and BAT^Δ/Δ mice.* *Pnpla2^fl/fl* [B6N.129S-Pnpla2^tm1Eek/J], *CX3CR1^gfp* [B6.129P-Cx3cr1^tm1Litt/J], and *TdTomato^fl/fl* [B6.Cg-Gt(ROSA) 26Sor^tm9(CAG-tdTomato)Hze/J] mice were crossed together and then crossed to *AdipoQ^creERT2* [C57BL/6-Tg(Adipoq-cre/ERT2)1Soff/J] or *Ucp1^Cre* [B6.FVB-Tg(Ucp1-cre)1Evdr/J] mice (all mice were purchased from The Jackson Laboratory; Ucp1^cre mice were kindly provided by Dr. Jean-François Tanti). *CCR2^creERT2* [C57BL/6NTac-Ccr2^tm2982(T2A-Cre7ESR1-T2A-mKate2)] mice were kindly provided by Dr. Burkhard Becher and crossed to *TdTomato^fl/fl* and *CCR2^GFP* [B6(C)-Ccr2^tm1.1Cln/J] provided by Dr. Marco Colonna[40,41]. For each experiment, co-housed littermate controls were used. Animal protocols were approved by the Institutional Animal Care and Use Committee of the French Ministry of Higher Education and Research and the Mediterranean Center of Molecular Medicine (INSERM U1065) and were undertaken in accordance with the European Guidelines for Care and Use of Experimental Animals. Animals had free access to food and water and were housed in a controlled environment with a 12 h light–dark cycle and constant temperature (20−23 °C).

*Induction of the Cre recombinase.* At 21 days-old (P21) mice were weaned and co-housed under a 12 h light/dark cycle. One week later, at P28, Adipo^Δ/Δ mice were injected every two days intra-peritoneally with either 10 mg/ml Tamoxifen (200μL per mouse in 10% EtoH and sunflower oil) or vehicle, for 1 week. Then, mice were allowed to rest for 2 weeks. Mice were then sacrificed, and tissues were harvested for further investigations. BAT^Δ/Δ mice were analyzed when 6–8-week-old.

**Conditional labeling of CCR2+ monocytes.** *CCR2^creERT2* *TdTomato* reporter mice were treated with tamoxifen dissolved in corn oil (20 mg/mL) by a single oral gavage (250 μL/mouse). Forty-eight hours later, animals were sacrificed and assessed for labeling efficiency in blood and BAT by flow cytometry.

**Monocyte depletion protocol.** Mice were injected every two days intra-peritoneally with 10 mg/ml Tamoxifen (200 μL per mouse in 10% EtoH and sunflower oil) for 1 week and were then allowed to rest for 1 week. Mice were then injected intra-peritoneally once a day for 5 days with 20 μg MC-21 (anti-CCR2) antibody or vehicle (PBS), and were sacrificed 16 h after the last injection for further investigations.

**In vivo Podoplanin blockade.** Mice were injected every two days intra-peritoneally with 10 mg/ml Tamoxifen (200 μL per mouse in 10% EtoH and sunflower oil) for 1 week and were then allowed to rest for 1 week. Mice were then injected intra-peritoneally with 100 μg anti-Podoplanin or isotype control once every two days for

6 days (three injections), and were sacrificed 16 h after the last injection for further investigations.

**Genotyping.** All primer sequences used for genotyping in this study are indicated in Supplementary Table 1.

DNA was extracted from tail biopsies by incubation with 50 mM NaOH for 30 min at 95 °C. DNA was amplified by PCR using DreamTaq Green PCR Master Mix (2X) (Thermo Scientific) and the following primers recognizing the sequences of the *AdipoQ*, *Pnpla2* (coding for ATGL), *TdTomato*, or *CX3CR1* gene.

PCR products were visualized on a 2% agarose gel.

Control mice present a single band 235 bp for *Pnpla2*, 410 bp for *CX3CR1*, 297 bp for *TdTomato*, and 514 bp for *CCR2*. Homozygotes mutant mice present a single band of 390 bp for *Pnpla2*, 500 bp for *CX3CR1*, 196 bp for *TdTomato*, 150 bp for *CCR2(GFP)*, 341 bp for *CCR2(creERT2)*. Heterozygote mice present each band of control and mutant genotype for these genes. The presence of the *AdipoQ*creERT2 allele is indicated by a 272 bp band, with an internal control 175 bp band. Control mice only present the internal control band.

### Metabolite analyses

*Glycerol, TG, and NEFA analysis.* Glycerol, TG, and NEFA contents were measured from serum. Free glycerol reagent and standard were used according to the manufacturer's protocol. NEFA-HR2 R1 + R2 FUJIFILM were used according to the manufacturer's protocol.

*Free fatty acids quantitation by NCI-GCMS.* Deuterated fatty acids were from Cayman (Bertin Pharma, Montigny le Bretonneux, France). Chemicals of the highest grade available were purchased from Sigma Aldrich (Saint-Quentin Fallavier, France). LCMSMS quality grade solvents were purchased from Fischer Scientific (Illkirch, France). Plasma was spiked with 5 µL of free fatty acids internal mix containing 635, 326, 95 ng, of linoleic acid d4, arachidonic acid-d8, DHA-d5, respectively. Plasma was mixed with 1.2 ml of Dole's reagent (Isopropanol/Hexane/Phosphoric acid 2 M 40/10/1 v/v/v). Free fatty acids were further extracted with 1 ml of Hexane and 1.5 ml of distilled water. Organic phase was collected and evaporated under vacuum. Fatty acids were analyzed as pentafluorobenzyl esters (PFB-FAs esters) by GCMS in negative chemical ionization mode. Calibration curves were obtained using linoleic acid (0.5−18 ng) arachidonic acid (0.3−11 ng) docosahexaenoic acid (0.1−3.5 ng) and docosapentaenoic acid (0.1−3.5 ng) extracted by the same method used for plasma. Linear regression was applied for calculations.

**Western blotting.** Tissues were harvested with a precellys (Bertin Instruments) in RIPA buffer. Tissues homogenate was then agitated for 1 h at 4 °C before being centrifuged at 14000 × *g* for 10 min at 4 °C. Supernatants were used for SDS-PAGE. Protein samples were resolved on 10% SDS-PAGE gels and were then transferred onto polyvinylidene difluoride (PVDF) membrane using a wet transfer system. Membranes were blocked in 5% (w/v) BSA in Tris-buffered saline-Tween (TBST) for one hour at room temperature. Membranes were incubated with primary antibody (HSP90 diluted in 4% BSA or ATGL diluted in 10% nonfat dry milk) overnight at 4 °C followed by the appropriate horseradish peroxidase-conjugated secondary antibody for 2 h at room temperature. Proteins were detected by ECL chemiluminescence (Pierce).

**Flow cytometry analysis.** All antibodies, supplier names, clones, and catalog numbers used in this study are provided in Supplementary Table 2. All antibodies were diluted according to the manufacturer's instructions.

*Blood analysis.* Just before the animal sacrifice, few drops of blood were collected by submandibular bleeding. Lysing buffer was used for red blood cells lysis. Cells were then centrifuged (400 × *g*, 5 min at 4 °C) and stained for 25 min protected from light in FACS buffer (RPMI medium, 0.3 mM EDTA, and 0.06% BSA). Cells were then washed, centrifuged and data were acquired on BD FACS Canto flow cytometer. Analysis was performed using FlowJo software (Tree Star).

*Tissue analysis.* Adipose tissues were harvested, shredded with scissors, and then incubated for 30 min with PBS containing 1.5 mg/ml collagenase A at 37 °C. Digested adipose tissue was homogenized using a 1 mL syringe with a 20G needle and passed through a 100 µm sieve. Spleens were crushed with a piston in PBS through a 100 µm sieve. The resulting suspension was lysed and processed as described in the previous paragraph.

Femurs and tibias were harvested and flushed with FACS buffer. Cells were then centrifuged (400 × *g*, 5 min at 4 °C), lysed, and stained as previously described.

*BrdU protocol.* Mice were injected with BrdU 16 h prior experiments. Blood and tissues were then processed and analyzed by flow cytometry as previously described.

**Histology.** Tissues were harvested and fixed in PFA-sucrose 30% 2 h at room temperature, then overnight at 4 °C. First, a paraffin infiltration was performed to dehydrated tissues (Myr Spin Tissue Processor STP 120). Then, tissues were embedded in paraffin (Center d'inclusion EC350-B&PMP). 7 µm sections were performed using a HM340E microtome (Microm Microtech, Francheville France) and immuno-stained as follow.

**Immunostaining.** Paraffin sections were de-paraffinized and H&E stained or rehydrated by washes in 100% xylene, 100% ethanol, 95% ethanol, 70% ethanol, and then washed in phosphate-buffered saline (PBS). Heat-induced antigen retrieval of sections was carried out using IHC Antigen retrieval solution (eBiosciences) and then washed in PBS. Sections were then blocked for 1 h in buffer (1% BSA, 1% Tween, in PBS), then incubated with primary antibodies overnight at 4 °C. Slides were then stained with DAPI (1 µg/mL) and secondary antibodies for 1 h at room temperature, and then washed thoroughly. Coverslips were mounted with ImmunoHistoMount. Following co-culture experiments, cells were fixed and incubated with PBS containing 1% BSA 1% Tween for 1 h. Cells were stained with DAPI and Texas Red Phalloidin for 1 h at room temperature, and then washed thoroughly. Coverslips were then mounted with ImmunoHistoMount. Immunostaining and histology data were analyzed using the Fiji-2 software[42].

**Co-culture experiments.** MEFs were cultured in DMEM medium containing 10% FBS, 4.5 g/L Glucose, 2 mM L-Glutamine, 1 mM Sodium Pyruvate, 50 U/mL Penicillin, 50 µg/mL Streptomycin. Monocytes were purified from the blood of C57BL/6 wild-type mice. Blood was collected by submandibular bleeding just before the animal sacrifice. Lysing buffer was used for red blood cells lysis. White blood cells were resuspended in FACS Buffer and stained with a depletion cocktail containing biotin-conjugated anti-CD3, anti-B220, anti-NK1.1, anti-Ly6G, anti-Ter119, and anti-SiglecF. Cells were then incubated with anti-biotin microbeads (Miltenyi Biotech) and passed through Miltenyi columns following the supplier's instructions. The flowthrough was collected and passed a total of four times on Miltenyi columns for optimal monocyte enrichment.

$3 \times 10^4$ MEFs were platted on glass coverslips in 24 well plates and left to adhere for 8 h before the addition of monocytes and/or anti-podoplanin. $1.5 \times 10^4$ monocytes were directly put in contact with MEFs or placed in cell culture inserts with 0.4 µm pores. Additionally, the podoplanin blockade was realized when indicated by adding 2 µg/mL anti-podoplanin (8.1.1 clone). Eighteen hours after addition of monocytes and/or anti-podoplanin, medium was removed, cells were washed once with PBS and fixed in 4% PFA for 20 min.

**Traction force microscopy.** Contractile forces exerted by MEF on 8 kPa hydrogels were assessed by traction force microscopy essentially as described (Liu et al. 2016). Briefly, polyacrylamide substrates with shear moduli of 8 kPa conjugated with fluorescent ed latex microspheres (0.5 µm, 505/515 ex/em) were purchased from Matrigen. MEF were plated on fluorescent bead-conjugated discrete stiffness gels and grown for 24 h, at which time they were treated with the indicated treatments for 18 h before traction force measurements. Images of gel surface-conjugated fluorescent beads were acquired for each cell before and after cell removal using a Axiovert 200 M motorized microscope stand (Zeiss) and a ×32 magnification objective. Tractions exerted by MEF were estimated by measuring bead displacement fields, computing corresponding traction fields using Fourier transform traction microscopy, and calculating root-mean-square traction using the PIV (particle Image velocity) and TFM (Traction force microscopy) package on ImageJ (Tseng et al. 2012). To measure baseline noise, the same procedure was performed on a cell-free region.

**Positron emission tomographic imaging/computed tomography (PET/CT) and post-PET biodistribution of ⁶⁴Cu-DOTA-ECL1i.** For PET/CT, 45−60 min dynamic scan was performed after the injection of ⁶⁴Cu-DOTA-ECL1i (3.7 MBq in 100 µL saline) via tail vein with Inveon PET/CT system (Siemens, Malvern, PA). The PET images were reconstructed with the maximum a posteriori algorithm and analyzed by Inveon Research Workplace. The organ uptake was calculated as the percent injected dose per gram of tissue in three-dimensional regions of interest without the correction for partial volume effect. Right after PET/CT, the mice were euthanized by cervical dislocation. Organs of interest were collected, weighed, and counted in a Beckman 8000 gamma counter (Beckman, Fullerton, CA). Standards were prepared and measured along with the samples to calculate the percentage of the injected dose per gram of tissue.

**Autoradiography.** After the collection of BATs, the radio activities in the tissues were detected by autoradiography using a Storm 840 Phosphorimager (GE, Marlborough, MA).

### Single-cell RNA-seq data analysis

*Cell preparation.* Adipose tissues were harvested, shredded with scissors, and then incubated for 30 min with PBS containing 1.5 mg/ml collagenase A at 37 °C.

Digested adipose tissues were homogenized using a 1 mL syringe with a 20G needle and passed through a 100 μm sieve. The resulting suspensions were centrifuged (400 × g, 5 min, 4 °C) and stained with anti-CD45-APC-Cy7 antibody. Cells were washed and passed again through a 100 μm sieve. Unstained cells were acquired to measure autofluorescence. Dapi was added to the cell preparation before the acquisition, and CD45$^+$ Dapi$^-$ cells were purified using a BD FACS Aria III cell sorter.

*Sequencing.* Cells were loaded on a Chromium Controller (10x Genomics) with a target output of 5000 cells per sample. Reverse transcription, cDNA synthesis/amplification, and library preparation were performed according to the 10x Genomics protocol (Chromium™ Single Cell 3′ Reagent Kit, v3.1 Chemistry). scRNA libraries were sequenced on an Illumina NextSeq 500/550 High Output flowcell: the forward read had a length of 28 bases that included the cell barcode and the UMI; the reverse read had a length of 55 bases that contained the cDNA insert.

Alignment, barcode assignment, and UMI counting with Cell Ranger v4.0.0 were used to perform sample demultiplexing, barcode processing, and single-cell 3′ counting. Cell Ranger's mkfastq function was used to demultiplex raw base call files from the HiSeq4000 sequencer into sample-specific FASTQ files.

Barcodes in both samples that were considered to represent noise and low-quality cells were filtered out using knee-inflection strategy available in DropletUtils[43] package (version 1.4.3). For analysis, Seurat package (version 3.1.0)[44] was used, genes which express in less than two cells and cells which have non-zero counts in less than 200 genes were additionally filtered from both barcode expression matrices, and the result matrices were used as analysis inputs. The fraction of mitochondrial genes was calculated for every cell, and cells with a mitochondrial fraction >2% were filtered out. After all filtering procedures, 2,242 cells were left in the scRNA-seq data of control sample, and 2,139 cells were left in scRNA-seq of Adipo$^{\Delta/\Delta}$ sample.

Both samples were normalized using *SCTransform* function with mitochondrial percentage as a variable to regress out in a second non-regularized linear regression. For integration purpose, variable features across the samples were selected by *SelectIntegrationFeatures* function with the number of features equal to 2000. Then the object was prepared for integration (*PrepSCTIntegration* function), the anchors were found (*FindIntegrationAnchors* function) and the samples were integrated into the whole object (*IntegrateData* function). The dimensionality of the object was reduced by principal component analysis, and the first 20 principal components (PCs) were used further to generate uniform manifold approximation and projection (UMAP) dimensionality reduction by *RunUMAP* function. Graph-based clustering was run using *FindNeighbors* and *FindClusters* with a resolution of 1.0 and the first 20 PCs as input, and the 17 clusters were identified. In order to exclude the technical bias across the samples, both the counts slot from *SCTransform* assay and the data slot from integrated assay were used as input for trajectory inference.

For visualization purposes, the custom labels were assigned to several clusters by merging multiple clusters for simplification (e.g., clusters 5, 6, and 14 were merged as T cells, clusters 4, 16 were merged as B cells and clusters 3, 15 were merged as NK cells). Violin plots were drawn using the *data* slot of *SCT* assay. To generate pathway enrichment plots we took the expression of genes from the pathway from the *data* slot of *SCT* counts assay, used standard normalization (z-score) for these vectors, and then calculated the average vector. The gene signature heatmap was drawn using the scaled *data* slot of the *integrated* assay.

For trajectory analysis, clusters assigned as monocytes and macrophages were used, and infer_trajectory function from the dyno package (version 0.1.2) was used with the available slingshot[45] singularity container (version 1.0.3). Trajectory visualization was implemented after dimensionality reduction by UMAP using dimred_umap function.

**Real-time qPCR**. All primer sequences used for RT-qPCR are indicated in Supplementary Table 1.

Total RNA was isolated using the RNeasy Plus Mini Kit and quantified using a Nanodrop. cDNA was prepared using 2,77 ng/μl total RNA by a RT-PCR using a high-capacity cDNA reverse transcription kit according to the manufacturer's instructions. Real-time qPCR was performed on cDNA using SYBR Green and GAPDH, ß-Actin, CXCL12, CCL2, or TNF-α forward and reverse primers (Invitrogen). qPCRs were performed on the StepOne device (Applied Biosystem). Results were normalized on GAPDH or ß-Actin gene expression. All conditions were performed in triplicates. All fold changes are expressed normalized to the untreated control.

**ELISA assays**. BATs were harvested and flash-frozen in liquid nitrogen. At the time of the experiment, tissues were homogenized in 500 μL sodium acetate solution (0.2 M, pH 4.5) using Precellys tissue homogenizer and centrifuged at 800 × g for 10 min at room temperature to pellet debris.

Mouse CXCL12 or CCL2 assays were performed on mice serum or tissue homogenates according to the recommended manufacturer instructions.

**Statistical analysis**. All the data of this paper are expressed in mean ± SEM. Mann−Whitney test was performed with GraphPad Prism 8 software to test the samples significance (*$p < 0.05$; **$p < 0.01$; ***$p < 0.001$; ****$p < 0.0001$).

**Reporting summary**. Further information on research design is available in the Nature Research Reporting Summary linked to this article.

## Data availability

Gene expression data (scRNA-seq) used in this study have been uploaded to the Gene Expression Omnibus (GEO) repository for public availability under the accession code GSE177635 and can be explored here https://artyomovlab.wustl.edu/scn/?token= GSE177635. All other data supporting the findings of this study are available from the corresponding author upon reasonable request. Source data are provided with this paper.

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

## Acknowledgements

We would like to thank Dr Pierre Leclère for the helpful discussions. We would like to thank Jean-Paul Pais de Barros from the Lipidomic Analytic platform (LAP) of the University of Burgundy for technical support. We thank the C3M Animal facility for technical support and the GIS-IBISA multi-sites platform Microscopie Imagerie Côte d'Azur (MICA), and particularly the imaging site of C3M (INSERM U1065) supported by Conseil Régional, Conseil Départemental, and IBISA. We sincerely thank Maéva Gesson and Marie Irondelle for their help. We acknowledge the flow cytometry facility from the «Institut de Pharmacologie Moléculaire et Cellulaire», part of the MICA GIS IBiSA labeled platform, and we thank Julie Cazareth for her help. We thank Dr. Jean-François Tanti for sharing Ucp1^Cre mice and Dr. Stephan Clavel for sharing MEFs. A.G. is supported by the French government, through the UCAJedi Investments in the Future projects managed by the National Research Agency (ANR) (ANR-15-IDEX-01). T.B. is supported by Agence Nationale de la Recherche (ANR-18-CE14-0025 and ANR-20-CE14-0006-02). M.F. and K.Z. were supported by the Government of Russian Federation (Grant 08-08). R.R.G. is supported by Center National de la Recherche Scientifique (CNRS). D.D. was supported by grants from the ANR and the European Union: EGID ANR-10-LABX-46. J.W.W. was supported by the National Institutes of Health (NIH) grant HL138163. L.Y.C. is supported by Institut National de la Sante et de la Recherche Medicale (INSERM), Fondation de France (00066474), and the European Research Council (ERC) consolidator program (ERC2016COG724838). S.I. is supported by Institut National de la Sante et de la Recherche Medicale (INSERM) and Agence Nationale de la Recherche (ANR-17-CE14-0017-01 and ANR-19-ECVD-0005-01).

## Author contributions

A.G., M.I.S., R.R.G., L.Y.C., and S.I. designed and performed experiments, and wrote the paper. A.G., M.I.S., J.M., N.K., N.V., J.G., and S.I. analyzed experiments. J.M., N.K., V.M., B.D., M.A., A.C., J.G., P.B., A.D., and N.V. helped performing experiments. A.J. and D.M. performed and analyzed the lipidomic analysis and edited the paper. H.P.L., D.S., and Y.L. performed and analyzed CCR2 PET/CT, biodistribution, and autoradiography experiments and edited the paper. M.M.F. and K.Z. analyzed scRNA-Seq data and edited the paper. T.B. performed and analyzed traction force microscopy experiments and edited the paper. B.B. provided CCR2^cre/ERT2 tdTomato mice and expertize on data analysis. J.W.W. performed monocyte fate-mapping experiments and edited the paper. M.M. provided MC-21 Ab, expertize on data analysis, and edited the paper. D.D. provided expertize on data analysis, wrote and edited the paper. A.G., M.I.S., S.I., M.M.F., K.Z., T.B., and Y.L. created figures. L.Y.C., R.R.G., and S.I. conceived the study. S.I. obtained funding for the project.

## Competing interests

The authors declare no competing interests.
