## [Peer Review File · Nature Communications]

REVIEWER COMMENTS

Reviewer #1 (Remarks to the Author):

In the manuscript, the authors study the diversity monocyte and its contribution to BAT functions during tissue remodeling/expansion. The manuscript is overall well written, and significance of the study objective is clear. Data presentation is good. Techniques and methods are very elaborative and appropriate.

The topic is interesting, but there are few concerns with this study.

1. CD11c monocytic population need to be focused also with regard to activated monocytic subset. Similarly, I would suggest the authors to consider/include the F4/80 macrophage population in their analysis.

2. CXCL14 is acritical BAT related chemokine relevant to macrophages recruitment. In my opinion authors should also check changes in the expression of CXCL14. Fractalkine-CX3CR1 interaction in recruitment of macrophages into the BAT which might contribute to adipose tissue remodeling and regulation of metabolic-related genes. It might be interested to look at this marker in the adipose tissue.

3. The authors are showing the reduction in the expression of M2 macrophage markers (CD206 & CD301) in the BAT of ADIPQ mice, it would be useful to also study concomitant changes in a few M1 markers in the same tissue samples to assure whether M1 markers expression remain unchanged or show modulation in their expression.

4. Authors found that TNF- α and CCL2 upregulated in BAT. It is reported well that TNF- α and CCL2 TNF- α inhibits UCP-1 expression in brown adipocytes. Please discuss this discrepancy.

Again my main concern is the data related to M1 and M2 markers and associated cytokines that should be included in the main figures.

Reviewer #2 (Remarks to the Author):

Brown Adipose Tissue (BAT) is responsible for non-shivering thermogenesis, and its activation has anti-obesity, anti-diabetic, and anti-dyslipidemia effects. Stunault and Gallerand et al. found that a previously unknown monocyte infiltration is required for BAT expansion. and proved it with elegant experiments. Overall, this manuscript is very well written and the experiments are beautifully executed. I strongly recommend its publication should the following points are improved.

Major points.

Physiological significance of monocyte infiltration. The authors use an artificial model of adipocyte-specific ATGL knockout, but does the same monocyte infiltration occur in physiological BAT expansion? e.g., 1 week of cold exposure at 4-5°C?

The most important experiment that proves the function of monocyte in BAT expansion is Figure 4, and more details of the depletion experiment should be written. The current method is too brief and could be explained independently in a protocol repository such as Protocol Exchange.

For example, in Figure 4A, at which time point did the authors count the monocyte by FACS? S4A should be included as a main figure. Also, should the control be PBS instead of isotype? For example, the authors used Isotype in Figure 5 (co-culture).

Several single-cell, nuclei RNA-sequencing of BAT have been published (e.g., GSE144720/PMID: 32597862), but monocyte has not been analyzed in detail. One would like to know if monocyte is also

present in these existing RNA-seq datasets.

Minor points.

There are no details on the CD45 sorts that were subjected to scRNA-seq. It should be detailed in the methods. Also, gates should be included as a supplement panel. Also, the gene names in Figure 1B are too small to read. It would be better to plot the expression levels of cluster-specific genes on UMAP rather than a heat map. In particular, the expression level of CD45 (PTPRC) should be included.

Kosaku Shinoda, Ph.D.

The Einstein-Mount Sinai Diabetes Research Center (ES-DRC)

REVIEWER COMMENTS

We would like to thank the reviewers for their careful reading of our manuscript, their positive feedback and their constructive comments. All the changes are highlighted in red in the revised version of the manuscript. We addressed each of their concerns below:

Reviewer #1 (Remarks to the Author):

In the manuscript, the authors study the diversity monocyte and its contribution to BAT functions during tissue remodeling/expansion. The manuscript is overall well written, and significance of the study objective is clear. Data presentation is good. Techniques and methods are very elaborative and appropriate. The topic is interesting, but there are few concerns with this study.

1. CD11c monocytic population need to be focused also with regard to activated monocytic subset. Similarly, I would suggest the authors to consider/include the F4/80 macrophage population in their analysis.

CD11c expression is higher on blood Ly6C^{low} monocytes when compared to Ly6C^{high} monocytes¹. The same expression pattern was observed on BAT monocytes with our scRNA-seq data (Reply Figure 1A). We did not observe any significant modulation in CD11c expression on BAT monocytes from control and Adipo^{ΔΔ} mice (Reply Figure 1A; now included in Figure 2J). However, we observed a significant increase in MHC II expression by BAT monocytes from Adipo^{ΔΔ} mice (Figure 2J). This result suggests enhanced monocyte activation, along with accelerated monocyte recruitment that was not paralleled by increased CD11c expression.

Although F4/80 was historically used to identify tissue macrophages², F4/80 expression was recently reported on eosinophils and dendritic cells (³ and Immgen.org). We observed consistent F4/80 (Adgre1) mRNA expression in BAT macrophages (clusters 5, 6, 7 and 8), but also lower (yet detectable) expression in neutrophils (cluster 0), dendritic cells (cluster 9) and monocytes (clusters 2, 3 and 4) (Reply Figure 1B). Macrophages can be identified across tissues through their co-expression of CD64 and MerTK, while monocytes are identified as CD64⁺ MerTK⁻ CD11b⁺ cells⁴. This gating strategy is shown in the Figure S2C and Reply Figure 1C. Monocyte-depleted animals allowed us to validate our monocyte gating strategy (Reply Figure 1D). In agreement with our scRNA-seq data, F4/80 expression was found predominantly on macrophages by flow cytometry (Reply Figure 1E). However, it was also found, at lower levels, on monocytes and CD11b⁺ CD64⁻ MerTK⁻ cells, making it hard to dissociate monocytes and macrophages using this marker (Reply Figure 1E).

Figure 1

(A) Single Cell RNA-seq analysis of CD11c (Itgax) mRNA expression by BAT myeloid cells and flow cytometry analysis of CD11c expression by BAT monocytes. (B) Single Cell RNA-seq analysis of F4/80 (Adgre1) mRNA expression by BAT myeloid cells. (C) Gating strategy used to identify BAT macrophages and monocytes. (D) Flow cytometry plot showing disappearance of BAT MerTK⁻ CD11b⁺ CD64⁺ monocytes in anti-CCR2-treated Adipo^{ΔΔ} mice. (E) Flow cytometry plot showing (left) F4/80⁺ cells (in red) among BAT CD45⁺ cells and (right) F4/80 expression by indicated BAT myeloid cells.

2. CXCL14 is a critical BAT related chemokine relevant to macrophages recruitment. In my opinion authors should also check changes in the expression of CXCL14. Fractalkine-CX3CR1 interaction in recruitment of macrophages into the BAT which might contribute to adipose tissue remodeling and regulation of metabolic-related genes. It might be interesting to look at this marker in the adipose tissue.

Cereijo et al. reported that BAT from CXCL14-deficient mice presents reduced numbers of F4/80⁺ cells, reduced thermogenic function and increased lipid content⁵. To apprehend how monocytes may interact with their local microenvironment in the expanding BAT of Adipo^{ΔΔ} mice, we recently performed bulk RNA-Seq analysis of BAT from vehicle-treated and MC-21-treated Adipo^{ΔΔ} mice. Principal Component Analysis revealed a distinct signature between the two groups (Reply Figure 2A). Interestingly, CXCL14 mRNA was found to be up-regulated in the expanding BAT in the absence of monocytes, when compared to control animals (Reply Figure 2A). Additional experiments are required to investigate how myeloid cells interact with BAT adipocytes through CXCL14-dependent and independent mechanisms in our experimental conditions, and these studies are beyond the scope of the current manuscript.

A recent report showed that both CX3CR1⁺ and CX3CR1⁻ macrophages are indeed present in mouse BAT at steady state⁶. The CX3CR1⁻ subset was found to be derived from CX3CR1⁺ cells that seed, at least in part, BAT-resident macrophages. These cells were proposed to control homeostatic thermogenesis in BAT⁶. Using CX3CR1^{GFP} reporter mice, we observed numerous CX3CR1⁺ cells among BAT immune cells (Reply Figure 2B). Around 25% of BAT macrophages were found to be CX3CR1⁺ at steady state and this proportion rose up to 40% in the BAT of Adipo^{ΔΔ} mice (Reply Figure 2C). The role of Fractalkine-CX3CR1 interactions in white adipose tissue biology remains controversial⁷⁻⁹. In this study, we used heterozygous CX3CR1^{GFP/+} reporter mice to track monocytes. Nevertheless, we got the opportunity to analyze CX3CR1-deficient (CX3CR1^{GFP/GFP}) mice during BAT expansion. Adipo^{ΔΔ} mice that were CX3CR1 heterozygous (GFP/+) and double knock-in (GFP/GFP) were found to have BAT weight, macrophage proportions and macrophage numbers comparable to CX3CR1^{+/+} Adipo^{ΔΔ} mice (Reply Figure 2D). This was added to the discussion in our revised manuscript (p7.I368.).

Figure 2

(A) PCA-plot and RNA-seq analysis of *Cxcl14* mRNA expression in BAT from vehicle-treated and MC-21-treated *Adipo^{ΔΔ}* mice. (B) Flow cytometry plot showing (left) CX3CR1-GFP⁺ cells (in green) among BAT CD45⁺ cells and (right) CX3CR1 expression by indicated BAT myeloid cells. (C) Proportions of CX3CR1-GFP⁺ BAT macrophages in Ctrl and *Adipo^{ΔΔ}* mice. (D) Analysis of BAT weight and macrophage content in CX3CR1^{+/+}, CX3CR1^{GFP/+} and CX3CR1^{GFP/GFP} *Adipo^{ΔΔ}* mice.

3. The authors are showing the reduction in the expression of M2 macrophage markers (CD206 & CD301) in the BAT of ADIPQ mice, it would be useful to also study concomitant changes in a few M1 markers in the same tissue samples to assure whether M1 markers expression remain unchanged or show modulation in their expression.

CD11c expression was reported to be an inflammatory marker in adipose tissue macrophages^{10,11}. CD11c expression was observed on Plin2^{high} macrophages (cluster 8) and matrix macrophages (cluster 5) in our scRNA-seq data (Reply Figure 3A). Flow cytometry analysis did not reveal any significant changes in membrane CD11c protein levels by BAT macrophages from control and *Adipo^{ΔΔ}* mice (Reply Figure 3A; now included in the revised Figure 2K). Expression of inflammatory cytokines by BAT macrophages appeared scarce in our scRNA-seq data, as illustrated by IL-1β and TNFα expression (Reply Figure 3B).

Figure 3

(A) (left) Single Cell RNA-seq analysis of CD11c (*Itgax*) mRNA expression by BAT myeloid cells and (right) flow cytometry analysis of CD11c expression by BAT macrophages. (B) Single Cell RNA-seq analysis of IL-1β and TNFα mRNA expression by BAT myeloid cells.

4. Authors found that TNF-α and CCL2 upregulated in BAT. It is reported well that TNF-α and CCL2 TNF-α inhibits UCP-1 expression in brown adipocytes. Please discuss this discrepancy. Again my main concern is the data related to M1 and M2 markers and associated cytokines that should be included in the main figures.

ATGL^{fl/fl} mice crossed with constitutive AdipoQ-driven Cre expression were shown to have reduced UCP1 expression in BAT and increased CCL2 and TNFα expression¹². We have now performed ELISA analysis of CCL2 and TNFα on BAT homogenates. ELISA analysis confirmed an increase in CCL2 protein levels in the BAT of *Adipo^{ΔΔ}* mice (Reply Figure 4A; now included in the revised Figure 3H). We did not detect differences in TNFα protein levels in the BAT of control and *Adipo^{ΔΔ}* mice. These results suggest that *Adipo^{ΔΔ}* mice display a very low-grade BAT inflammation with specific CCL2-mediated monocyte recruitment during BAT expansion. The potential role of CCL2 and TNFα during thermogenesis is discussed in the revised version of our manuscript (p8.1390.).

Previous reports demonstrated that UCP1^{cre}-mediated BAT-specific ATGL deficiency did not impact BAT UCP1 mRNA and protein levels and did not affect thermogenesis at room temperature^{13,14}. We generated and analyzed BAT from UCP1^{cre} x ATGL^{fl/fl} mice, and we observed an accumulation of macrophages and monocytes similar to what we observed in the BAT of *Adipo^{ΔΔ}* mice (Reply Figure 4B; now included in the revised Supplemental Figure 2F).

We measured a decrease in CD206 and CD301 mean fluorescence intensity (MFI) when considering total BAT macrophages obtained from control and *Adipo^{ΔΔ}* mice. However, our scRNA-seq data showed the presence of multiple macrophage subsets, and did not reflect changes in CD206 and CD301 expression in M2-like macrophages (cluster 6) and matrix macrophages (cluster 5), the main macrophage subsets in healthy BAT. Slingshot trajectory analysis suggested that BAT monocytes can differentiate into CD206^{low} CD301^{low} macrophages (clusters 7 and 8) that accumulate in the BAT of *Adipo^{ΔΔ}* mice in comparison to controls (Reply Figure 4C). Flow cytometry analysis allowed us to validate the accumulation of CD206^{low} macrophages in the BAT of *Adipo^{ΔΔ}* mice while CD206⁺ macrophages did not down-regulate this marker (Reply Figure 4D). We therefore conclude that the decrease in CD206 and CD301 MFI observed on total BAT macrophages reflects accumulation of CD206^{low} CD301^{low} monocyte-derived macrophages rather than a shift in their polarization.

Figure 4

(A) Quantification of CCL2 and TNF α protein levels in BAT homogenates from control (n=8) and Adipo Δ/Δ (n=7) mice by ELISA. (B) Quantification of BAT monocyte and macrophage numbers in control (n=6) and BAT Δ/Δ (n=4) mice using flow cytometry. (C) Single Cell RNA-seq analysis of CD206 (Mrc1, left) and CD301 (Clec10a, right) expression by BAT myeloid cells represented in a differentiation pattern predicted by Slingshot analysis. (D) Flow cytometry plots representing CD206 expression by BAT macrophages from Ctrl and Adipo Δ/Δ mice.

Reviewer #2 (Remarks to the Author):

Brown Adipose Tissue (BAT) is responsible for non-shivering thermogenesis, and its activation has anti-obesity, anti-diabetic, and anti-dyslipidemia effects. Stunault and Gallerand et al. found that a previously unknown monocyte infiltration is required for BAT expansion and proved it with elegant experiments. Overall, this manuscript is very well written and the experiments are beautifully executed. I strongly recommend its publication should the following points are improved.

Major points.

Physiological significance of monocyte infiltration. The authors use an artificial model of adipocyte-specific ATGL knockout, but does the same monocyte infiltration occur in physiological BAT expansion? e.g., 1 week of cold exposure at 4-5°C?

Using the same CCR2-tracing PET-CT approach as in the current study, we previously reported that mice housed at thermoneutrality (30°C) show reduced BAT monocyte content in comparison to mice housed at 22°C¹⁵. Thermoneutrality induces BAT expansion through TG accumulation, which happens in the absence of thermogenic activity. This result could suggest that BAT activation may induce monocyte recruitment to this tissue, or that thermoneutrality represses the constant influx of monocytes that we observed at steady state. We recently performed bulk RNA-seq analysis of BAT from vehicle-treated and MC-21-treated Adipo^{ΔΔ} mice housed at 22°C. Pathway enrichment analysis revealed differential expression of pathways involved in heat production, further suggesting that monocytes play a key role in BAT activation (Reply Figure 5).

We have been working on animal housing at different ambient temperatures¹⁵ and those experimental set-ups require dedicated rooms with limited user access, adapted light/dark cycles and controlled environment. We currently, do not have access to such a facility in our local structures. We would be able to perform overnight (16h) experiment, but this experiment would not correctly address the role of monocytes during thermogenesis because short-term 4°C housing triggers a stress-related inflammation. Nevertheless, we believe that this is an important question to address in our future directions.

pathway	pval	padj	log2err	ES	NES	size
REACTOME : RESPIRATORY ELECTRON TRANSPORT, ATP SYNTHESIS BY CHEMIOSMOTIC COUPLING AND HEAT PRODUCTION BY UNCOUPLING PROTEINS	1.39688607171146e-20	2.67981564809908e-18	1.16906996676109	0.722954964549412	3.16429674102401	76

Figure 5

Pathway enrichment analysis showing modulations of the Reactome heat production pathway in BAT from vehicle-treated and MC-21-treated Adipo^{ΔΔ} mice.

The most important experiment that proves the function of monocyte in BAT expansion is Figure 4, and more details of the depletion experiment should be written. The current method is too brief and could be explained independently in a protocol repository such as Protocol Exchange.

We have added more details on the depletion experiment (p6.l296.).

For example, in Figure 4A, at which time point did the authors count the monocyte by FACS? S4A should be included as a main figure. Also, should the control be PBS instead of isotype? For example, the authors used Isotype in Figure 5 (co-culture).

We have included panel S4A in the main figures (Figure 4A). For each monocyte depletion experiment, mice were sacrificed 16 hours after the MC-21 injection. Blood and tissues were analyzed at the moment of sacrifice.

MC-21-mediated monocyte depletion has been extensively described¹⁶, and no non-specific or Fc receptor-mediated effects have been reported with this treatment^{17,18}. MC-21 treatment leads to a rapid (4-6 hours) monocyte disappearance from the blood circulation, while other leukocytes remain unaffected. Blood profiling allowed us to confirm that MC-21 treatment induces a specific monocyte depletion in Adipo^{ΔΔ} mice (Reply Figure 6). We did not deem the use of an isotype control necessary considering the low dose used (20μg) and short treatment period.

Figure 6
(Left) Gating strategy and (Right) quantification of blood Neutrophils, T cells and B cells from Ctrl and Adipo^{Δ/Δ} mice.

Several single-cell, nuclei RNA-sequencing of BAT have been published (e.g., GSE144720/PMID: 32597862), but monocyte has not been analyzed in detail. One would like to know if monocyte is also present in these existing RNA-seq datasets.

We would like to thank the reviewer for this suggestion. We discussed these previous studies in the discussion section of our manuscript (p7.1358.). These snRNA-sequencing experiments were performed on whole BAT and present few CD45⁺ cells, making it hard to have a robust identification of monocytes and macrophages. To our knowledge, we report the first scRNA-seq of BAT immune cells (CD45⁺ cells).

Minor points.

There are no details on the CD45 sorts that were subjected to scRNA-seq. It should be detailed in the methods. Also, gates should be included as a supplement panel. Also, the gene names in Figure 1B are too small to read. It would be better to plot the expression levels of cluster-specific genes on UMAP rather than a heat map. In particular, the expression level of CD45 (PTPRC) should be included.

We have added more details in the methods about the cell preparation and sorting strategy used for scRNA-seq (p29.1997). Since macrophages display inherent autofluorescence, we acquired part of the sample before addition of DAPI to position the sorting gate. CD45⁺ Dapi⁺ cells were sorted to perform scRNA-seq (Reply Figure 7A; now included in the revised Figure S1A). We have addressed the concerns raised by Reviewer 2 concerning the presentation of the scRNA-seq data. Gene names in are now bigger on the heatmap, and expression levels of cluster-specific genes have been shown on UMAP (now included in the revised Figure 1B). The expression level of Ptpcr (Reply Figure 7B; now included in the revised Figure S1A) has been included in the supplemental Figure 1.

Figure 7

(A) Sorting strategy used to isolate BAT CD45⁺ cells for scRNA-seq. (B) Analysis of Ptprc expression among BAT cells that were submitted to scRNA-sequencing.

- 1 Ingersoll, M. A. *et al.* Comparison of gene expression profiles between human and mouse monocyte subsets. *Blood* **115**, e10-19, doi:10.1182/blood-2009-07-235028 (2010).
- 2 Hume, D. A., Robinson, A. P., MacPherson, G. G. & Gordon, S. The mononuclear phagocyte system of the mouse defined by immunohistochemical localization of antigen F4/80. Relationship between macrophages, Langerhans cells, reticular cells, and dendritic cells in lymphoid and hematopoietic organs. *J. Exp. Med.* **158**, 1522-1536, doi:10.1084/jem.158.5.1522 (1983).
- 3 Ginhoux, F. *et al.* The origin and development of nonlymphoid tissue CD103⁺ DCs. *J. Exp. Med.* **206**, 3115-3130, doi:10.1084/jem.20091756 (2009).
- 4 Gautier, E. L. *et al.* Gene-expression profiles and transcriptional regulatory pathways that underlie the identity and diversity of mouse tissue macrophages. *Nat. Immunol.* **13**, 1118-1128, doi:10.1038/ni.2419 (2012).
- 5 Cereijo, R. *et al.* CXCL14, a Brown Adipokine that Mediates Brown-Fat-to-Macrophage Communication in Thermogenic Adaptation. *Cell Metab.* **28**, 750-763 e756, doi:10.1016/j.cmet.2018.07.015 (2018).
- 6 Wolf, Y. *et al.* Brown-adipose-tissue macrophages control tissue innervation and homeostatic energy expenditure. *Nat. Immunol.* **18**, 665-674, doi:10.1038/ni.3746 (2017).
- 7 Nagashimada, M. *et al.* CX3CL1-CX3CR1 signalling deficiency exacerbates obesity-induced inflammation and insulin resistance in male mice. *Endocrinology*, doi:10.1210/endo/bqab064 (2021).
- 8 Polyak, A. *et al.* The fractalkine/Cx3CR1 system is implicated in the development of metabolic visceral adipose tissue inflammation in obesity. *Brain. Behav. Immun.* **38**, 25-35, doi:10.1016/j.bbi.2014.01.010 (2014).
- 9 Morris, D. L., Oatmen, K. E., Wang, T., DelProposto, J. L. & Lumeng, C. N. CX3CR1 deficiency does not influence trafficking of adipose tissue macrophages in mice with diet-induced obesity. *Obesity (Silver Spring)* **20**, 1189-1199, doi:10.1038/oby.2012.7 (2012).
- 10 Li, P. *et al.* Functional heterogeneity of CD11c-positive adipose tissue macrophages in diet-induced obese mice. *J. Biol. Chem.* **285**, 15333-15345, doi:10.1074/jbc.M110.100263 (2010).
- 11 Xu, X. *et al.* Obesity activates a program of lysosomal-dependent lipid metabolism in adipose tissue macrophages independently of classic activation. *Cell Metab.* **18**, 816-830, doi:10.1016/j.cmet.2013.11.001 (2013).
- 12 Schoiswohl, G. *et al.* Impact of Reduced ATGL-Mediated Adipocyte Lipolysis on Obesity-Associated Insulin Resistance and Inflammation in Male Mice. *Endocrinology* **156**, 3610-3624, doi:10.1210/en.2015-1322 (2015).
- 13 Schreiber, R. *et al.* Cold-Induced Thermogenesis Depends on ATGL-Mediated Lipolysis in Cardiac Muscle, but Not Brown Adipose Tissue. *Cell Metab.* **26**, 753-763 e757, doi:10.1016/j.cmet.2017.09.004 (2017).

- 14 Shin, H. *et al.* Lipolysis in Brown Adipocytes Is Not Essential for Cold-Induced Thermogenesis in Mice. *Cell Metab.* **26**, 764-777 e765, doi:10.1016/j.cmet.2017.09.002 (2017).
- 15 Williams, J. W. *et al.* Thermoneutrality but Not UCP1 Deficiency Suppresses Monocyte Mobilization Into Blood. *Circ. Res.* **121**, 662-676, doi:10.1161/CIRCRESAHA.117.311519 (2017).
- 16 Mack, M. *et al.* Expression and characterization of the chemokine receptors CCR2 and CCR5 in mice. *J. Immunol.* **166**, 4697-4704, doi:10.4049/jimmunol.166.7.4697 (2001).
- 17 Giladi, A. *et al.* Cxcl10(+) monocytes define a pathogenic subset in the central nervous system during autoimmune neuroinflammation. *Nat. Immunol.* **21**, 525-534, doi:10.1038/s41590-020-0661-1 (2020).
- 18 Winkler, E. S. *et al.* Human neutralizing antibodies against SARS-CoV-2 require intact Fc effector functions for optimal therapeutic protection. *Cell* **184**, 1804-1820 e1816, doi:10.1016/j.cell.2021.02.026 (2021).

REVIEWERS' COMMENTS

Reviewer #1 (Remarks to the Author):

Thank you for addressing all the comments- Excellent study-

Reviewer #2 (Remarks to the Author):

The authors responded appropriately to the reviewers' points, and I think this is a great paper that deserves to be published in Nature Communications.

--

Kosaku Shinoda, Ph.D.

Assistant Professor

Albert Einstein College of Medicine

The Einstein-Mount Sinai Diabetes Research Center (ES-DRC)

Nathan Shock Center for excellence on Biology of Aging Research (E-NSC)

1301 Morris Park Avenue Room: 355

Bronx, NY 10461

Tel: 718.678.1189 (office)

Tel: 718.678.1258 (lab)

kosaku.shinoda@einsteinmed.org

Reviewer #1 (Remarks to the Author):

Thank you for addressing all the comments- Excellent study-

Reviewer #2 (Remarks to the Author):

The authors responded appropriately to the reviewers' points, and I think this is a great paper that deserves to be published in Nature Communications.

We would like to thank the reviewers for their constructive and positive evaluations of our manuscript.